# Receptor-targeted engineered probiotics mitigate lethal Listeria infection

Rishi Drolia [1,2], Mary Anne Roshni Amalaradjou[1,3], Valerie Ryan[1], Shivendra Tenguria[1,2], Dongqi Liu[1,2], Xingjian Bai[1], Luping Xu[1], Atul K. Singh[1], Abigail D. Cox [4], Victor Bernal-Crespo [4], James A. Schaber[5], Bruce M. Applegate[1,6], Ramesh Vemulapalli [4,7] & Arun K. Bhunia [1,2,4,6 ✉]

Probiotic bacteria reduce the intestinal colonization of pathogens. Yet, their use in preventing fatal infection caused by foodborne *Listeria monocytogenes* (*Lm*), is inconsistent. Here, we bioengineered *Lactobacillus* probiotics (BLP) to express the *Listeria* adhesion protein (LAP) from a non-pathogenic *Listeria* (*L. innocua*) and a pathogenic *Listeria* (*Lm*) on the surface of *Lactobacillus casei*. The BLP strains colonize the intestine, reduce *Lm* mucosal colonization and systemic dissemination, and protect mice from lethal infection. The BLP competitively excludes *Lm* by occupying the surface presented LAP receptor, heat shock protein 60 and ameliorates the *Lm*-induced intestinal barrier dysfunction by blocking the nuclear factor-κB and myosin light chain kinase-mediated redistribution of the major epithelial junctional proteins. Additionally, the BLP increases intestinal immunomodulatory functions by recruiting FOXP3+T cells, CD11c+ dendritic cells and natural killer cells. Engineering a probiotic strain with an adhesion protein from a non-pathogenic bacterium provides a new paradigm to exclude pathogens and amplify their inherent health benefits.

[1] Molecular Food Microbiology Laboratory, Department of Food Science, Purdue University, West Lafayette, IN, USA. [2] Purdue Institute of Inflammation, Immunology and Infectious Disease, Purdue University, West Lafayette, IN, USA. [3] Department of Animal Science, University of Connecticut, Storrs, CT, USA. [4] Department of Comparative Pathobiology, Purdue University, West Lafayette, IN, USA. [5] Bindley Bioscience Research Center, Purdue University, West Lafayette, IN, USA. [6] Purdue University Interdisciplinary Life Science Program, Purdue University, West Lafayette, IN, USA. [7] Department of Veterinary Pathobiology, Texas A&M University, College Station, TX, USA. ✉email: bhunia@purdue.edu

The intestinal mucosa is the first site for the dynamic interaction of enteric pathogens with the host[1,2]. Commensal microbiota play an essential role in protecting host intestines from exogenous pathogen infections[3]. However, several enteric pathogens have evolved strategies to escape from commensal-mediated colonization resistance[4]. Probiotics are live commensal microbes that confer antagonism against intestinal pathogen and health benefit when consumed in adequate amounts[5,6]. The commonly used probiotics belong to strains/ species of lactobacilli and bifidobacteria, which are natural inhabitants of the intestine[7]. While the precise mechanism of action of probiotics that confer a benefit to the host is unknown, it is proposed that probiotics compete with pathogens for adhesion sites[8], improve microbial balance[9], restore epithelial barrier function[7,10,11], and enhance the epithelial immune response[12,13], thus averting infection and the consequent pathology.

The major limitations of probiotics for prophylactic or therapeutic use are their poor ability to colonize the intestine and consequently exhibit a weaker capacity to compete with pathogens and exert immunomodulatory actions[7,14,15]. To overcome these limitations of traditional probiotics, next-generation bioengineered probiotic strains can be designed to incorporate desirable traits[16–18]. One such strategy is to bioengineer a probiotic strain that expresses a pathogen-specific surface protein to preferentially bind the host receptor[19–22]. This enables a more targeted approach that blocks the interaction of the pathogen with the host[15,18,21].

The foodborne pathogen, Listeria monocytogenes (Lm) crosses the intestinal barrier to cause fatal systemic infections (case fatality rate is 20–30%) in newborns, the elderly, and other immunocompromised individuals[23]. Lm can also cross the blood–brain barrier and cause meningitis and encephalitis, as well as the placental barrier, resulting in abortion or stillbirth in pregnant women[24]. Currently, there is no vaccine against listeriosis. The general preventive precautionary guidelines outlined by the Centers for Disease Control and Prevention (CDC) are thorough cooking of meat, safe food handling practices, and avoidance of the Food and Drug Administration (FDA) designated high-risk foods, such as frankfurters, soft cheeses made with unpasteurized milk, paté, smoked fish, and cantaloupe. Therefore, cost-effective strategies that prevent Lm infection and the progression of the disease are urgently needed.

In the genus, Listeria, Lm, and L. ivanovii are pathogenic while L. innocua (Lin) and 14 other Listeria species are nonpathogenic[25]. Lm is well adapted to survive in the harsh environment of the intestine[26,27] and overcomes gut-associated innate defense[28] to cross the intestinal epithelial barrier. The M-cells overlying Peyer's patches[29,30], the Listeria adhesion protein (LAP)[31,32], and the bacterial invasion protein, Internalin A (InlA)[33]-mediated pathways are important for Lm to cross the host intestinal barrier[34]. While the M-cell pathway is used by many enteric pathogens, the InlA and LAP-mediated pathways are highly specific to Lm. The InlA accesses its cognate host cell basolateral receptor, E-cadherin, during epithelial cell extrusion[35] and goblet cell exocytosis[33] which propels internalization of luminal Lm by transcytosis. The LAP interacts with its cognate host cell receptor, heat shock protein 60 (Hsp60) at the apical side and causes epithelial barrier dysfunction that promotes Lm translocation across the epithelial barrier[31].

LAP is a housekeeping alcohol acetaldehyde dehydrogenase (lmo1634) present in both pathogenic and nonpathogenic Listeria species[36]. However, LAP exhibits virulent attributes only in pathogenic Listeria because of a lack of secretion and surface reassociation of LAP on nonpathogenic species of Listeria[36,37]. The interaction of LAP with its host receptor Hsp60 leads to activation and nuclear translocation of nuclear factor-κB (NF-κB), which also results in the activation of myosin light-chain kinase

(MLCK)[31]. The activated MLCK phosphorylates myosin light chain (MLC) for cellular redistribution of the tight junction (TJ) proteins; claudin-1 and occludin, and the adherens junction (AJ) protein; E-cadherin, leading to cell–cell junctional opening[31]. Consequently, Lm executes efficient translocation across the intestinal barrier by manipulating the LAP–Hsp60–NF-κB–MLCK axis[34].

We previously showed that a recombinant Lactobacillus paracasei engineered to express the Lm LAP reduces the interaction of Lm in vitro[38]. However, the demonstration of the in vivo functionality and the molecular basis of protection of such engineered probiotic strain is lacking. Furthermore, the expression of a protein from a pathogenic bacterium (Lm) may raise health or regulatory concern of such an engineered probiotic strain.

Here, we expressed the LAP from a nonpathogenic Listeria (Listeria innocua) on the surface of a Lactobacillus casei strain (a more commonly used probiotic strain with proven immunomodulatory actions)[39–41]. At the same time, we also expressed the Lm LAP on the surface of L. casei. Remarkably, the bioengineered Lactobacillus probiotic (BLP) strains robustly colonize the intestine, dramatically reduce mucosal Lm colonization and extraintestinal dissemination and protect mice from lethal infection. We further demonstrate that the BLP occupies the host receptor, Hsp60, and prevents Lm translocation and infection by competitive exclusion, the Lm-induced NF-κB and MLCK activation, MLC phosphorylation, and subsequent redistribution of the major epithelial junctional proteins (claudin-1, occludin, and E-cadherin) to preserve intestinal epithelial barrier integrity. Additionally, BLP augments immunomodulatory action through recruiting intestinal FOXP3+T cells, CD11c+ dendritic cells, and natural killer (NK) cells. This approach of engineering a probiotic strain with an adhesion protein from a nonpathogenic bacterium represents a unique strategy to exclude pathogens and amplify the inherent health benefits associated with probiotics.

## Results

**BLP strains expressing LAP from Lin (non-pathogen) or Lm prevent Lm adhesion to Caco-2 and MDCK cells.** LAP from Lm shares a 99.4% amino acid sequence identity with that from Lin[36,42] (Fig. 1a and Supplementary Fig. 1a). In pathogenic Listeria species, LAP is secreted and re-associated on the cell surface of the bacterium[36]. However, in nonpathogenic species of Listeria such as Lin, LAP fails to be secreted extracellularly and reassociated on its cell surface[36,37] and thus, cannot mediate the adhesion of Lin to epithelial cells. The high identity in the amino acid sequence of LAP from Lm and Lin prompted us to investigate whether the LAP from Lin exhibits a similar adhesion function to that of LAP from Lm. For this purpose, we cloned and expressed the lap gene of Lin strain F4248, in the lap-deficient Lm strain (lap−). Immunoblotting of the cell wall and secreted proteins confirmed the surface expression and secretion of LAP in the homologous (lap−+lapLm) and the heterologous (lap−+lapLin) complemented lap-deficient Lm strain (Fig. 1b). In line with our previous observations[36,37], LAP was not detected in the secreted or cell wall fraction of Lin (Fig. 1b). The heterologous complementation (lap−+lapLin) restored the ability of the lap− strain to adhere (Fig. 1c), invade (Fig. 1d), and translocate (Fig. 1e) across the human enterocyte-like Caco-2 cell monolayers to levels similar to those of the wild-type (WT) Lm strain (F4244, serovar 4b, clinical strain) or the homologous complemented lap-deficient Lm strain (lap−+lapLm). These data suggest that the LAP from Lin is functionally similar to the LAP from Lm.

The LAP from Lm and Lin exhibited similar adhesion functions; therefore, we cloned the lap ORF (2.6 kb) from both

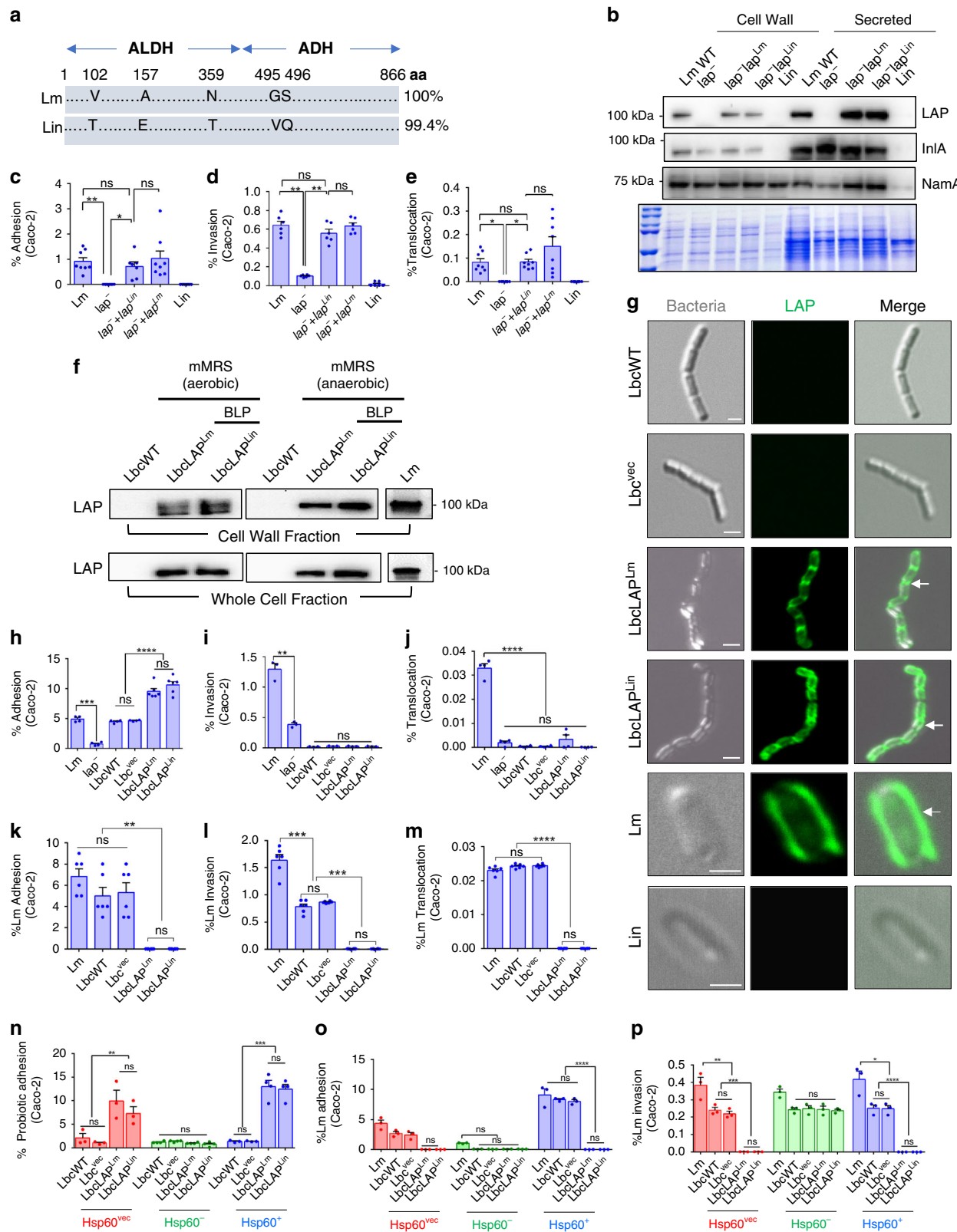

*Lm* and *Lin* separately into the *Lactobacillus* expression vector, pLP401T. Next, we expressed LAP on the surface of the vancomycin-resistant *Lactobacillus casei* wild-type ATCC334 (LbcWT) strain via the cell wall anchoring protein, protease P (PrtP)[38,43]. We selected the vancomycin-resistant strain (300 μg/ml) by sequentially culturing the bacterium in increasing concentrations of antibiotic to precisely enumerate this

strain from intestinal samples during in vivo studies. The resulting BLP strains were designated LbcLAP^Lm and LbcLAP^Lin, respectively. The BLP strains showed a similar growth rate as the LbcWT strain in deMan Rogosa Sharpe (MRS) and modified MRS broths (Supplementary Fig. 1b). The cell surface expression of LAP on both the BLP strains grown in mMRS (to induce the expression of the heterologous protein) was

**Fig. 1 BLP strains expressing LAP from *Listeria innocua* (*Lin*) or *Lm* prevent *Lm* interaction in vitro. a** Schematics showing 99.4% amino acid sequence similarity of LAP from *Lm* and *Lin* (NCBI database). **b** Immunoblot showing cell wall expression and secretion of LAP in $lap^-+lap^{Lm}$ and $lap^-+lap^{Lin}$ strains. InlA and NamA: fractionation marker controls. Coomassie-stained gel (bottom panel) showing equal loading. **c–e** Restoration of adhesion (**c**, MOI 10, 1 h, $n = 8$ for all groups except for $lap^-+lap^{Lin}$, $n = 7$), invasion (**d**, MOI 10, 2 h, $n = 6$), and translocation (**e**, MOI 10, 2 h, $n = 8$) of the $lap^-+lap^{Lin}$ strain in Caco-2 cells. **f** Immunoblot showing LAP expression in BLP strains in the cell wall and whole-cell fractions. **g** Immunofluorescence micrographs showing cell surface expression of LAP in BLP strains expressing LAP of *Lm* (arrows) or *Lin* (arrows). *Lm* (arrow) but not *Lin* shows the surface expression of LAP. Scale, 1 μm. **h–j** Increased adhesion of the BLP strains (MOE 10) (**h**, 1 h, $n = 4, 4, 6, 6, 4, 4$, for each group, respectively) but not invasion (**i**, 2 h, $n = 3$) and translocation (**j**, 2 h, $n = 4$) in Caco-2 cells. **k–m** Increased inhibition of *Lm* (MOI 50) adhesion (**k**, $n = 6$), invasion (**l**, $n = 6$), and translocation (**m**, $n = 6$) by the BLP strains (MOE 10, 24 h) in Caco-2 cells. **n** Increased adhesion of BLP (MOE 10, 1 h) strains in Hsp60$^{Vec}$ cells ($n = 3$) and Hsp60$^+$ cells ($n = 4$), but not in Hsp60$^-$ cells ($n = 3$-4) Caco-2 cells. **o, p** Increased inhibition of *Lm* adhesion (**o**, $n = 3$) and invasion (**p**, $n = 3$) by the BLP strains in Hsp60$^{Vec}$ and Hsp60$^+$, but not in Hsp60$^-$ Caco-2 cells. Panels **c–e** and **h–p** represent the mean ± SEM of $n =$ independent cell culture wells from three independent experiments. The one-way (**c–e** and **h–m**) or two-way (**n–p**) ANOVA with Tukey's multiple comparisons was used. For all analyses, ****$P < 0.0001$; ***$P < 0.001$; **$P < 0.01$; *$P < 0.05$; ns no significance. Panels **b** and **g** are representative of three independent experiments. Source data are provided as a Source Data file.

confirmed by immunoblotting (Fig. 1f) and immunofluorescence staining (Fig. 1g).

Next, we examined the adhesion characteristics of BLP strains with Caco-2 cell monolayers. Relative to the LbcWT strain, the BLP strains showed significantly increased adhesion (approximately twofold increase) to Caco-2 cells following 1 h exposure (Fig. 1h) and remained adhered at a significantly higher level (approximately threefold) following 24 h after exposure (Supplementary Fig. 1c). Surprisingly, similar to the LbcWT strain, the BLP stains showed negligible invasion to (Fig. 1i) or translocation (Fig. 1j) across the Caco-2 monolayers. In contrast, *Lm*-WT strain as a positive control displayed a significantly higher invasion and translocation (Fig. 1i, j). These data suggest that the expression of LAP by the BLP strains allows them to adhere at a significantly higher level to epithelial cells but does not afford them the ability to invade or translocate.

To test whether the BLP strains prevent *Lm* interaction with epithelial cells, we exposed the LbcWT or the BLP strains for 24 h before the *Lm* challenge. Relative to the LbcWT, pretreatment of BLP strains significantly reduced (~90–99% reduction) *Lm* adhesion (Fig. 1k and Supplementary Fig. 1d), invasion (Fig. 1l and Supplementary Fig. 1e), and translocation (Fig. 1m and Supplementary Fig. 1f) across Caco-2 and MDCK cells (less-permissive to *Lm*)[35]. Furthermore, the ability of the BLP strain to prevent *Lm* adhesion, invasion, and translocation (Supplementary Fig. 1g–i) in Caco-2 cells was significantly more evident for the *Lm*-WT strain than those exposed to the $lap^-$ strain. These data suggest that the *Lm* LAP interaction with intestinal epithelial cells is crucial for BLP-mediated exclusion of *Lm*. Of note, pretreatment of Caco-2 or MDCK cells with a control Lbc strain harboring a pLP401T vector without the *lap* insert (Lbc$^{Vec}$) showed a limited reduction of *Lm* adhesion as the LbcWT. These data thus dismiss any extraneous contribution by the virgin plasmid. The anti-adhesion effect of the BLP strains was also not due to any bactericidal compounds as the agar well diffusion assay ruled out the production of any bacteriocin-like inhibitory substances by the LbcWT or the BLP strains (Supplementary Fig. 1j).

The host receptor for LAP is the mammalian chaperone protein, Hsp60[44,45]. To determine whether the expression of LAP in the BLP strains prevented *Lm* adhesion by interfering or blocking the *Lm* LAP–Hsp60 interactions, we assessed the adhesion profiles of the BLP and their ability to prevent *Lm* interaction in shRNA-mediated Hsp60 knocked-down (Hsp60$^-$, ~70%) or plasmid-mediated Hsp60 overexpressed (Hsp60$^+$, ~60%) Caco-2 cells (Supplementary Fig. 1k). Relative to LbcWT strains, the BLP strains showed significantly higher adhesion (~4-fold increase) to the non-targeting shRNA vector-control Caco-2 cells (Hsp60$^{Vec}$) that express basal levels of Hsp60 (Fig. 1n and

Supplementary Fig. 1k). In contrast, the adhesion of the BLP strains in Hsp60$^-$ Caco-2 cells was significantly reduced to levels similar to those of the LbcWT strains. Conversely, the adhesion of BLP strains was significantly more pronounced (approximately ninefold increase) in Hsp60$^+$ cells. Consistent with the adhesion profiles of the BLP, their concomitant ability to prevent *Lm* adhesion, invasion, and translocation was observed in Hsp60$^{Vec}$ cells and these effects were significantly more evident in Hsp60$^+$ cells (Fig. 1o, p and Supplementary Fig. 1l). In contrast, the BLP strains showed limited inhibition of *Lm* adhesion (Fig. 1o), invasion (Fig. 1p), and translocation (Supplementary Fig. 1l) as the LbcWT in Hsp60$^-$ cells. To further confirm the contribution of host Hsp60 in *Lm* interaction with intestinal epithelial cells, we pretreated Caco-2 cells with an Hsp60-specific antibody before *Lm* exposure, which significantly reduced the adhesion (approximately tenfold), invasion (approximately threefold), and translocation (~12-fold) of the *Lm*-WT strain (Supplementary Fig. 1m–o). Collectively, these data demonstrate that the LAP–Hsp60 interaction is critical for BLP-mediated increased adhesion and their ability to exclude *Lm*.

**Oral administration of BLP prevents *Lm* lethal infection in vivo.** Before performing in vivo mouse experiments, we first verified the survival of all *L. casei* strains in simulated gastric fluid (SGF) and simulated intestinal fluid I (SIF-I) and II (SIF-II) by plate counting (Supplementary Fig. 2a–c). We also confirmed the expression of LAP in the BLP strains following sequential exposure to SGF, SIF-I, and SIF-II by Western blotting (Supplementary Fig. 2d).

Next, we determined the impact of oral treatment of BLP on *Lm* translocation across the intestinal barrier and systemic dissemination in A/J mice that are highly sensitive to oral *Lm* challenge[31,37,46,47]. We supplied freshly grown LbcWT or BLP strains (4–8 × 10⁹ colony-forming units, CFU/ml) daily in autoclaved drinking water for 10 days before oral challenge with *Lm*-WT strain (5 × 10⁸ CFU/mouse) (Fig. 2a). Following 10 days of *L. casei* treatment, the BLP strains showed significantly increased (approximately tenfold) colonization in the intestine (Fig. 2b) and more specifically to the mucosa of the ileum, cecum, and colon (Supplementary Fig. 2e) compared to the LbcWT strain. Remarkably, the increased colonization of the BLP strains in the intestine was maintained despite *Lm* infection (Fig. 2b). The BLP strains sustained the expression of LAP as confirmed by immunoblotting of fecal isolates from mice that were treated with these strains (Supplementary Fig. 2f). Furthermore, lactobacilli were not detected intracellularly in the intestine (gentamicin-resistant CFU) or the extraintestinal sites (Supplementary Fig. 2g) of *L. casei*-treated mice, suggesting the inability of BLP strains to cross the intestinal barrier. Collectively, these data indicate that

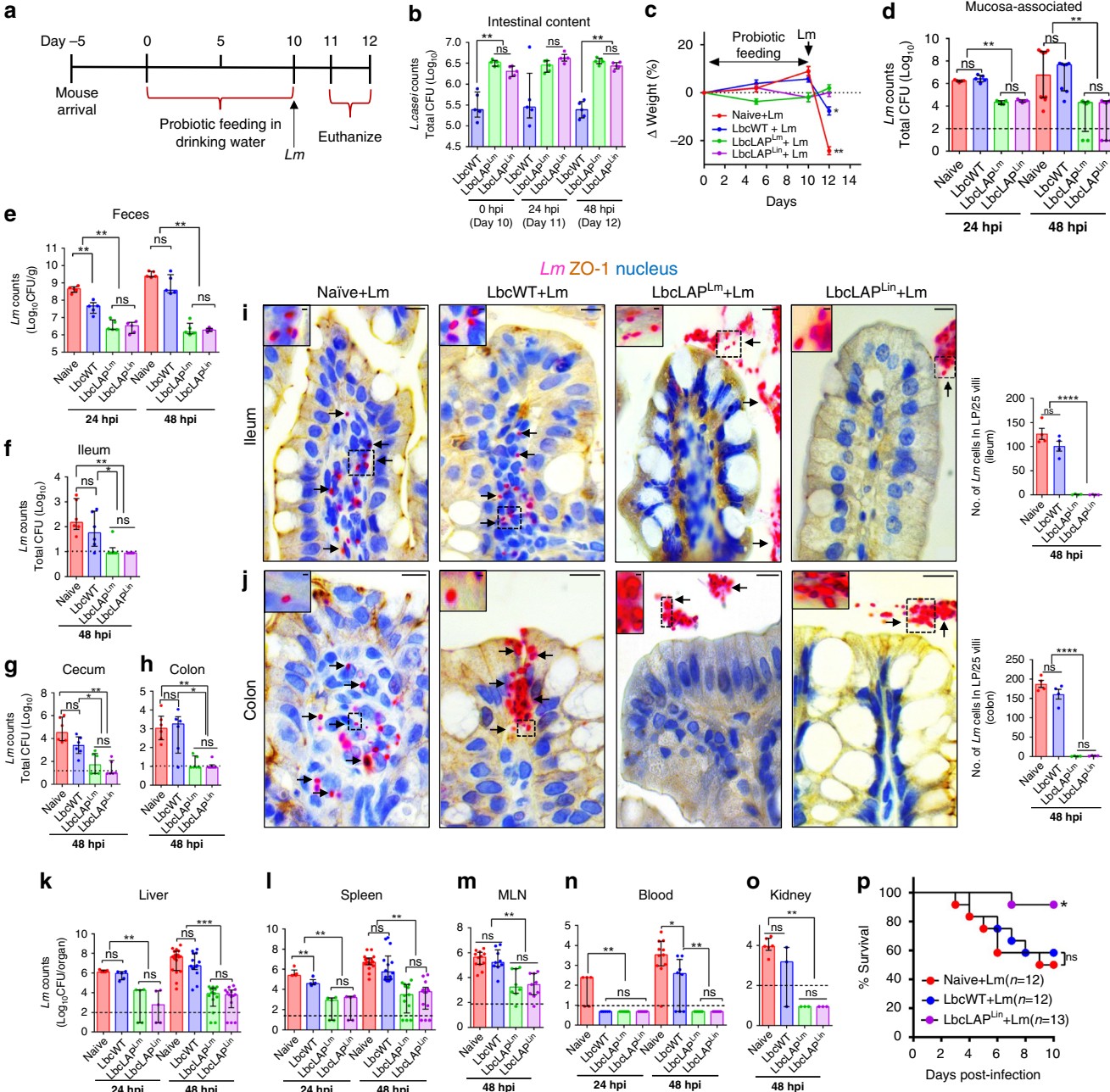

**Fig. 2 BLP prevents lethal *L. monocytogenes* infection in mice. a** Schematics showing mouse experiment protocol. **b** Increased BLP counts in the intestinal content of mice (*n* = 5) on days 10, 11, and 12. **c** Normalized mouse body weight (mean ± SD, *n* = 5) on days 0, 5, 10, and 12. **d–h** Reduced *Lm* burdens in the intestine (**d**, mucosa-associated, *n* = 5 at 24 hpi; and *n* = 10, 8, 8, 8, at 48 hpi for each group, respectively), shed in the feces (**e**, *n* = 5) and intracellular location in the ileum (**f**, *n* = 6), cecum (**g**, *n* = 6), and colon (**h**, *n* = 6). **i, j** Micrographs of ileal (**i**) and colonic villi (**j**) immunostained for ZO-1 (brown) and *Lm* (red, arrows) and counterstained for the nucleus (blue) at 48 hpi. Bars, 10 μm. The boxed areas were enlarged. Bars, 1 μm. *Lm* counts (mean ± SEM) in ileal or colonic lamina propria (LP; right panels). Dots represent an average of 25 villi from a single mouse, four mice/group, *n* = 100 villi. *Lm* is observed in the LP (arrows) in naïve or LbcWT-treated mice, but confined in the lumen (arrows) in BLP-treated mice. **k–o** Reduced *Lm* burdens in the liver (**k**), spleen (**l**), (**k** and **l**, *n* = 5 at 24 hpi; and *n* = 17, 14, 13, 14, at 48 hpi), MLN (**m**, *n* = 11, 9, 8, 9, for each group, respectively), blood (**n**, *n* = 5 at 24 hpi; and *n* = 11, 8, 8, 8, at 48 hpi for each group, respectively) and kidney (**o**, *n* = 6, 3, 3, 3, for each group, respectively). **p** Increased survival of BLP-treated mice at LD₅₀ dose. *$P < 0.05$ Kaplan–Meier log-rank test. For plots **b**, **d–h**, and **k–o** each point represents an individual mouse (*n*) from three to six independent experiments (median and interquartile range). Dashed horizontal lines indicate the limit of detection for each organ/tissue. Mann–Whitney nonparametric (two-tailed) test (**b**, **d–h**, **k–o**) was used and comparisons were made between each treatment group individually or by using the one-way (**c**, **i**, and **j**) ANOVA test followed by a Tukey's multiple comparisons. For all analyses, ****$P < 0.0001$; ***$P < 0.001$; **$P < 0.01$; *$P < 0.05$; ns no significance. Source data are provided as a Source Data file.

LAP promotes increased intestinal colonization and mucosal adhesion of the BLP strains in vivo.

When we evaluated the health indices of these groups of mice following the *Lm* challenge, the BLP-treated groups appeared healthy. In contrast, naive (mock-treated) or LbcWT-treated mice that were challenged with *Lm* appeared ill, displayed ruffled fur, labored breathing, recumbency, restricted movement, non-responsiveness to external stimuli (Supplementary Movies 1–5), and lost ~5–20% body weight (Fig. 2c) as was evident with the increased clinical sign scores of these mice at 48 h post infection (hpi) (Supplementary Fig. 2h). Consistent with these observations, the burdens of *Lm* adhered to the intestine were significantly lower (~100-fold reduction) in BLP-treated mice compared to LbcWT-treated mice at 24 and 48 hpi (Fig. 2d). Fecal shedding data revealed that a vast majority of *Lm* were shed in the feces of BLP-treated mice at 12 hpi (Supplementary Fig. 2i), while at 24 and 48 hpi, the numbers of *Lm* in the feces were significantly lower (~600-fold reduction) in BLP-treated mice, relative to LbcWT-treated mice (Fig. 2e and Supplementary Fig. 2i). Similarly, the burdens of *Lm* that invaded the ileum, cecum, and colon (gentamicin-resistant CFU) were significantly lower (~100-fold reduction) in BLP-treated mice than that of the LbcWT-treated mice (Fig. 2f–h). These differences in the mucosal invasion were also evident morphologically, as immunostaining confirmed significantly increased *Lm* penetration into the ileal (Fig. 2i) and colonic (Fig. 2j) lamina propria in naive or LbcWT-treated mice that were infected with *Lm*. In contrast in BLP-treated mice, *Lm* cells were restricted to the luminal surface of the epithelium.

In line with impaired translocation of *Lm* across the intestine in BLP-treated mice, the dissemination of *Lm* was significantly reduced to the liver, spleen, and MLN (100 to 1000-fold reduction) compared to LbcWT-treated mice in each organ/tissue at 24 or 48 hpi (Fig. 2k–m). Furthermore, *Lm* was undetectable in the kidneys and blood of BLP-treated mice (Fig. 2n, o).

To determine, if the decreased *Lm* burdens in the BLP-treated mice impacted their survival, we monitored the LbcLAP$^{Lin}$-treated (as both the BLP strains; LbcLAP$^{Lm}$ and LbcLAP$^{Lin}$, showed a similar reduction in *Lm* tissue burdens) mice for survival rates following *Lm* challenge at LD$_{50}$ ($2.5 \times 10^9$ CFU/mouse). Ten days post infection, ~92% of LbcLAP$^{Lin}$-treated mice while only 60% of LbcWT-treated mice survived (Fig. 2p). Taken together, these data unequivocally demonstrate that oral administration of BLP strains prevents *Lm* translocation across the intestinal barrier and the consequent fatal systemic infection.

**BLP colonizes and persists in the intestine and limits *Lm* translocation, despite discontinuous administration**. Next, we determined the persistence of the BLP strains during the 10 days treatment period (Fig. 3a, supplied in drinking water) and 10 days post treatment by monitoring fecal shedding. Relative to the LbcWT strain, the BLP strain (LbcLAP$^{Lin}$) was consistently recovered at significantly higher levels than the LbcWT in the feces during the 10 days treatment period (Fig. 3b, ~200% increase on day 2 and ~1000% increase on day 10). Once *L. casei* treatment was stopped, the fecal shedding of the LbcWT strain gradually decreased and dropped below the detection limit (i.e., 130 CFU/g feces) from day 14 to day 20 (Fig. 3b). In sharp contrast, the BLP strain was recovered at significantly higher levels in the fecal samples (Fig. 3b, ~2700% increase on day 12 and ~16,000% increase on day 16) over time and was stably maintained at ~$1 \times 10^{4.5}$ CFU/g until day 20 (Fig. 3b). These data suggest that BLP, but not LbcWT was able to colonize and persist,

albeit at a reduced level, in the gastrointestinal tract even 10 days after probiotic treatment was stopped.

We next investigated if BLP persistence (after probiotic treatment was stopped) would confer protection against *Lm* infection. Post *L. casei* treatment, mice were challenged on days 10, 15, 20 with *Lm* and sacrificed 48 hpi i.e., on days 12, 17, and 22, respectively (Fig. 3a), and *L. casei* colonization and *Lm* burdens in the intestinal and extraintestinal tissues were analyzed. Relative to the LbcWT strain, significantly increased colonization of the BLP strain was observed in the intestine on days 12, 17, and 22 (Fig. 3c, ~21,000%, ~7000%, and ~28,000%, respectively). Consequently, the *Lm* counts that invaded the ileum, cecum, and colon (gentamicin-resistant CFU) were significantly lower in BLP-treated mice than that of the naive (mock-treated) or LbcWT-treated mice (Fig. 3d–f ~99% reduction on day 12, and ~90–95% reduction on days 17 and 22). Similarly, the dissemination of *Lm* was significantly reduced to the liver, spleen, and MLN compared to naive- or LbcWT-treated mice in each organ/tissue (Fig. 3g-i, ~95% reduction on days 12 and 17, and ~90% reduction on day 22) and significantly fewer *Lm* were found in the blood (Fig. 3j, ~99% reduction on days 12, 17, and 22). Collectively, these data suggest that the highest BLP-mediated protection was observed on days 12 and 17 i.e., 2–7 days after the BLP treatment was stopped and the protective effect was slightly diminished 12 days after treatment i.e., until the end of the trial (day 22).

**BLP displays increased co-aggregation with *Lm* and competitively excludes *Lm* by occupying the epithelial Hsp60 receptor**. We previously demonstrated that secreted LAP has a strong affinity for the cell wall of *Lm*. However, LAP does not re-associate with the cell wall of *Lin* (Fig. 1b, g)[36,37]. To better understand the mechanism that allows the BLP strains to prevent *Lm* adhesion, we hypothesized that the expression of LAP in the BLP may promote the attachment of the BLP strains to the *Lm* cell wall to form co-aggregates (attachment to bacterial cells of different species). To test this hypothesis, we co-incubated suspensions containing equal CFU's (1:1 ratio) of LbcWT or BLP cells with *Lm* or *Lin* cells and used *Listeria*-specific immuno-magnetic beads (IMB) to capture *Lm* and *Lin* cells in co-incubated bacterial cell suspensions (Fig. 4a). The bacterial cells attached to the IMB were plated on *Listeria* selective MOX or *Lactobacillus* selective MRS agar plates. While the number of *Lm* and *Lin* cells captured by the IMB was comparable among LbcWT or BLP co-incubated suspensions, strikingly, the number of co-captured BLP cells was significantly higher (~100-fold increase) compared to LbcWT cells only in the presence of *Lm* cells (Fig. 4a). In contrast, the co-captured LbcWT and BLP counts were similar in the presence of *Lin* which is possibly a result of the low affinity of LAP (expressed on the cell wall of BLP) to re-associate with the surface of *Lin*. In the absence of *Lm* or *Lin*, IMB had negligible interaction with *L. casei* cells. These observations were further confirmed microscopically, where we observed significantly increased (~80–90%) co-aggregates of the BLP strains with the IMB captured *Lm* cells expressing the green fluorescence protein (*Lm*-GFP, Fig. 4b, arrows and Fig. 4c), relative to LbcWT cells. Furthermore, pretreatment of BLP with anti-LAP mAb to block the surface LAP reduced the co-capture levels of BLP by the IMB-*Lm* complex to levels similar to those of LbcWT strains, while an isotype IgG control had a negligible effect confirming the involvement of LAP in the formation of BLP-*Lm* co-aggregates (Supplementary Fig. 3a). Collectively, these data provide direct evidence that LAP promotes the co-aggregation of the BLP strains with *Lm* cells.

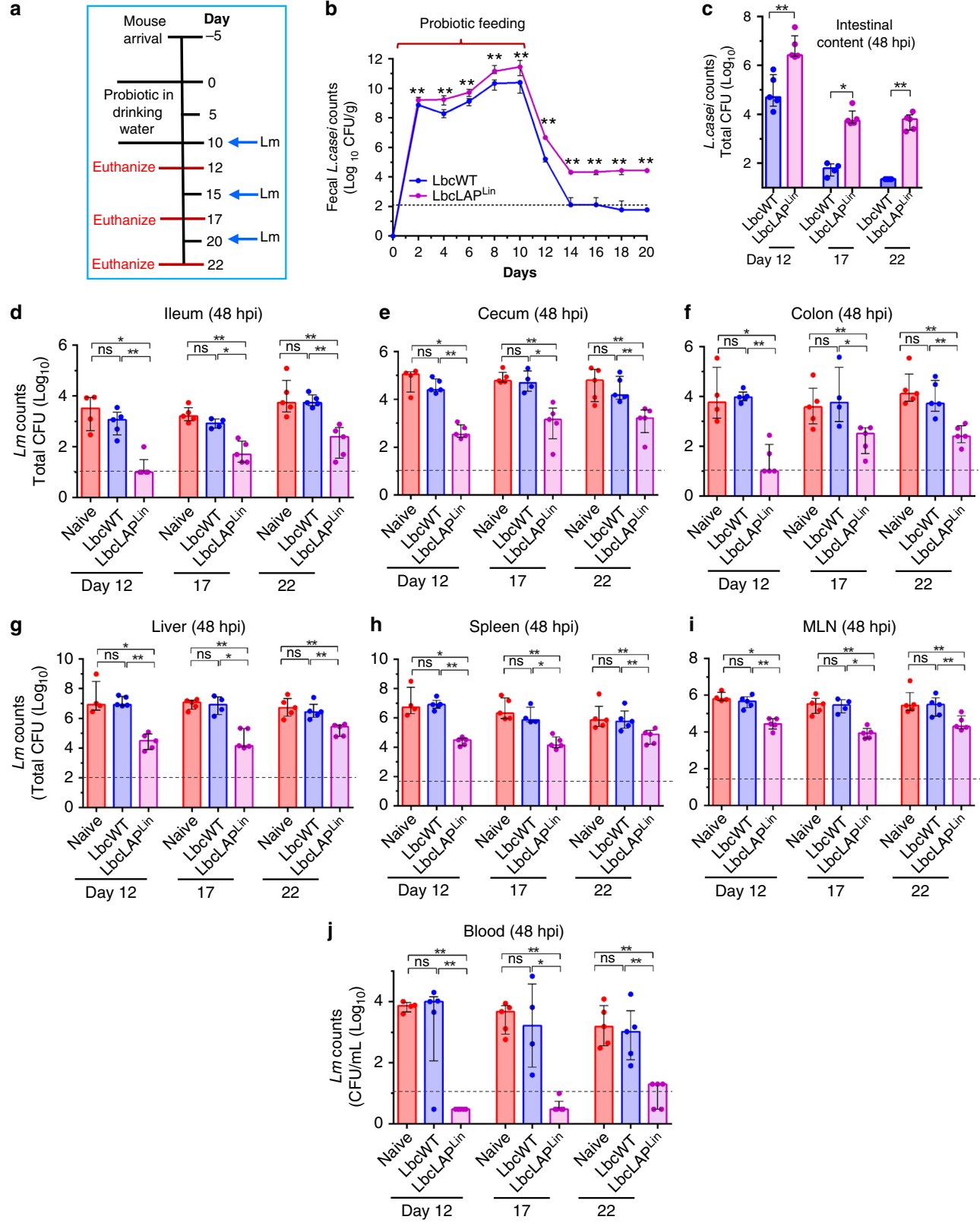

Next, we investigated whether the BLP strains inhibited *Lm* adhesion to Caco-2 cells by blocking *Lm*'s ability to adhere via competitive exclusion by co-incubating two cultures at 1:1 ratio. Relative to the LbcWT or Lbc^Vec strain, the BLP strains showed markedly increased (approximately twofold) adhesion and significantly lowered (~95%) the attachment of *Lm* to Caco-2

cells (Fig. 4d). Additionally, the BLP strains remained adhered to Caco-2 cells at significantly higher levels (approximately fivefold), despite *Lm* exposure. Analysis of cell wall proteins (an equivalent amount from both *Lm* and BLP strains) revealed 50–60% increased levels of surface LAP in BLP strains than that of the *Lm* strain (Supplementary Fig. 3b). Thus, the enhanced

**Fig. 3 BLP colonization and persistence in the intestine limits *Lm* translocation, despite discontinuous administration. a** Schematics showing mouse experiment protocol: mice were treated with *L. casei* (LbcWT) or BLP (LbcLAP$^{Lin}$) strain supplied in drinking water replenished daily (4–8 × 10$^9$ CFU/ml) for 10 days (0–9 days) and then challenged with *Lm* F4244 (~5 × 10$^8$ CFU/animal) on days 10, 15, and 20. **b** Increased fecal shedding ($n = 5$) of BLP strain (LbcLAP$^{Lin}$) than the LbcWT over time. **c** Increased BLP counts in the intestinal (duodenum–colon) content of BLP-treated mice on days 12 ($n = 4, 5, 5$, for each group, respectively), 17 ($n = 5, 4, 5$, for each group, respectively), and 22 ($n = 5$) following *Lm* challenge. **d–j** BLP-mediated reduced *Lm* burdens at 48 hpi in the intracellular location in the ileum (**d**), cecum (**e**), and colon (**f**) (**d–f**, gentamicin-resistant CFU), and in the liver (**g**), spleen (**h**), mesenteric-lymph node (MLN, **i**), and blood (**j**) in mice on days 12 ($n = 4, 5, 5$, for each group, respectively), 17 ($n = 4, 5, 5$, for each group, respectively), and 22 ($n = 5$). For plots **c–j**, each point represents an individual mouse ($n$) from one experiment. Bar and brackets represent the median and interquartile range, respectively for the data points in each group. Dashed horizontal lines indicate the limit of detection for each organ/tissue. Mann–Whitney nonparametric test (two-tailed, **b–j**) was used and comparisons were made between each treatment group individually. For all analyses, **$P < 0.01$; *$P < 0.05$; ns no significance. Source data are provided as a Source Data file.

expression of LAP on the BLP cell wall possibly enabled BLP strains to competitively exclude the adhesion of *Lm* cells.

We next hypothesized that the LAP-expressing BLP strains occupy the epithelial cell membrane expressed Hsp60 receptor to competitively exclude *Lm*. We first confirmed membrane localization of Hsp60 by immunostaining Caco-2 cells and consistent with our previous report[32], we observed co-localization of Hsp60 with the peripheral membrane protein; ZO-1 (Supplementary Fig. 3c). Next, we examined whether the BLP strains interact with Hsp60 receptors by immunostaining the BLP with an anti-LAP mAb. Both the BLP strains co-localized with the surface-expressed Hsp60 (localized with ZO-1) in mono-incubated (BLP alone, ~10 BLP cells/field, Fig. 4e, f, Supplementary Fig. 3d) or co-incubated (BLP + *Lm*-GFP at 1:1 ratio, ~7 BLP cells/field, Fig. 4g, h, Supplementary Fig. 3d) Caco-2 cells. Furthermore, upon co-incubation, the majority of *Lm* cells were competitively excluded or co-aggregated with the BLP strains (Fig. 4g, h and Supplementary Fig. 3e). Similarly, Giemsa staining of co-incubated (BLP+*Lm*) Caco-2 cells depicted the competitive exclusion of *Lm* cells (Supplementary Fig. 3f). In strong contrast, microscopic analysis of *Lm*-GFP cells and Giemsa staining (Supplementary Fig. 3e, f) showed markedly higher *Lm* adhesion in LbcWT+*Lm* co-incubated Caco-2 cells. Together, these results suggest that BLP strains form co-aggregates with *Lm* and occupy the surface-expressed Hsp60 receptor-binding sites to competitively exclude *Lm*.

Likewise, immunostaining of the colonic tissues of *Lm*-challenged mice confirmed the co-aggregation of both the BLP strains (probed with anti-LAP mAb) with *Lm* (probed with anti-*Lm* pAb) on the brush border of the colonic villi and the absence of *Lm* cells in the lamina propria (Fig. 5a and Supplementary Fig. 4, middle and right panels). In contrast, in LbcWT-treated mice, *Lm* cells were abundant in the lamina propria (Fig. 5a and Supplementary Fig. 4, left panels) consistent with our earlier findings (Fig. 2j). Furthermore, the BLP strains formed biofilm-like structures on the brush border of the colonic surface and restricted *Lm* cells at the epithelial surface (Fig. 5a and Supplementary Fig. 4, middle and right panels). To further validate these in vivo observations, we measured the biofilm formation of LbcWT and BLP strains in microtiter plates by crystal violet staining. Relative to the LbcWT strain, the BLP strains showed a significant increase in biofilm production (Fig. 5b) in monoculture (BLP alone, approximately threefold) or in co-culture with *Lm* (BLP + *Lm* 1:1 ratio, ~3.5-fold). However, no significant difference in biofilm formation was observed between LbcWT and BLP stains when co-cultured with another gastroenteric pathogen, *Salmonella enterica* serovar Typhimurium (Fig. 5b), demonstrating that the binding of LAP (expressed on BLP) was unique and highly specific to the *Lm* cell wall. Additionally, fluorescent in-situ hybridization (FISH) using an *L. casei*-specific 16s rDNA probe also confirmed the presence of Lbc aggregates on the surface of epithelial cells in the

BLP-treated mice while in LbcWT-treated mice, Lbc were mostly restricted to the luminal mucus layer (Fig. 5c, d). Collectively, these data confirm that the BLP strains adhere to the epithelial cells of the intestinal mucosa, co-aggregate with *Lm* cells in the intestinal lumen and at the lumen–epithelial interphase, and competitively exclude the interaction of *Lm* with the host intestinal epithelial cells.

**BLP prevents *Lm* from causing intestinal barrier loss by maintaining mucus-producing goblet cells and limiting epithelial apoptotic and proliferative cells.** *Lm* crosses the intestinal villus epithelium during goblet cell (GC) exocytosis[33] and epithelial cell extrusion[35] and upon infection, *Lm* accelerates intestinal villus epithelium proliferation while decreasing the number of GCs[48]. However, the decrease in GCs is detrimental for the host since it reduces the thickness of the protective mucosal barrier[48]. Thus, we next analyzed the effect of the treatment of BLP on the *Lm*-induced changes in intestinal histopathology, mucus-producing GCs, and epithelial cell proliferation and apoptosis.

Histopathological analyses of the ileal tissues identified loss of apical villus epithelial cells and significantly increased numbers of polymorphonuclear and mononuclear cells infiltrating the lamina propria in *Lm*-challenged naive or LbcWT-treated mice at 48 hpi (Fig. 6a, b). In contrast, ileal tissues of BLP-treated mice had significantly reduced signs of inflammation and displayed similar histological characteristics of naive uninfected mice. Of note, treatment of mice with *L. casei* strains (LbcWT or the BLP) alone did not cause histopathological changes, relative to naive uninfected mice (Fig. 6a, b).

GC counts following Alcian blue staining (Supplementary Fig. 5a) and MUC2 (the major component of mucin[49])-positive GC counts after immunostaining (Fig. 6c, d), showed a significant increase (~30% and ~25%, respectively) in pre-challenged BLP-treated mice compared to naive or LbcWT-treated mice. At 48 h post *Lm* challenge, the number of GCs and MUC2$^+$ GCs markedly decreased (~35% and ~40%, respectively) in the ileal tissues of naive or LbcWT-treated mice. In sharp contrast, the number of GCs (Supplementary Fig. 5a) and MUC2$^+$ GC's (Fig. 6c, d) were maintained in the ileal tissues of BLP-treated mice at 48 hpi, relative to naive uninfected mice.

Next, we measured epithelial proliferative and apoptotic responses by immunostaining of Ki67$^+$ cycling cells and cleaved caspase-3 (CC3$^+$) apoptotic cells. Relative to naive uninfected mice, the ileal tissues of naive or LbcWT-treated mice at 48 hpi displayed markedly increased Ki67$^+$ (~50%, Fig. 6e, f and Supplementary Fig. 5b) and CC3$^+$ (~400%, Fig. 6g, h) cells. In contrast, the ileal tissues of the BLP-treated mice displayed similar numbers of Ki67$^+$ and CC3$^+$ cells at 48 hpi as the naive uninfected mice.

Together, these results suggest that treatment with BLP but not the LbcWT strain promotes mucus-producing GCs, maintains

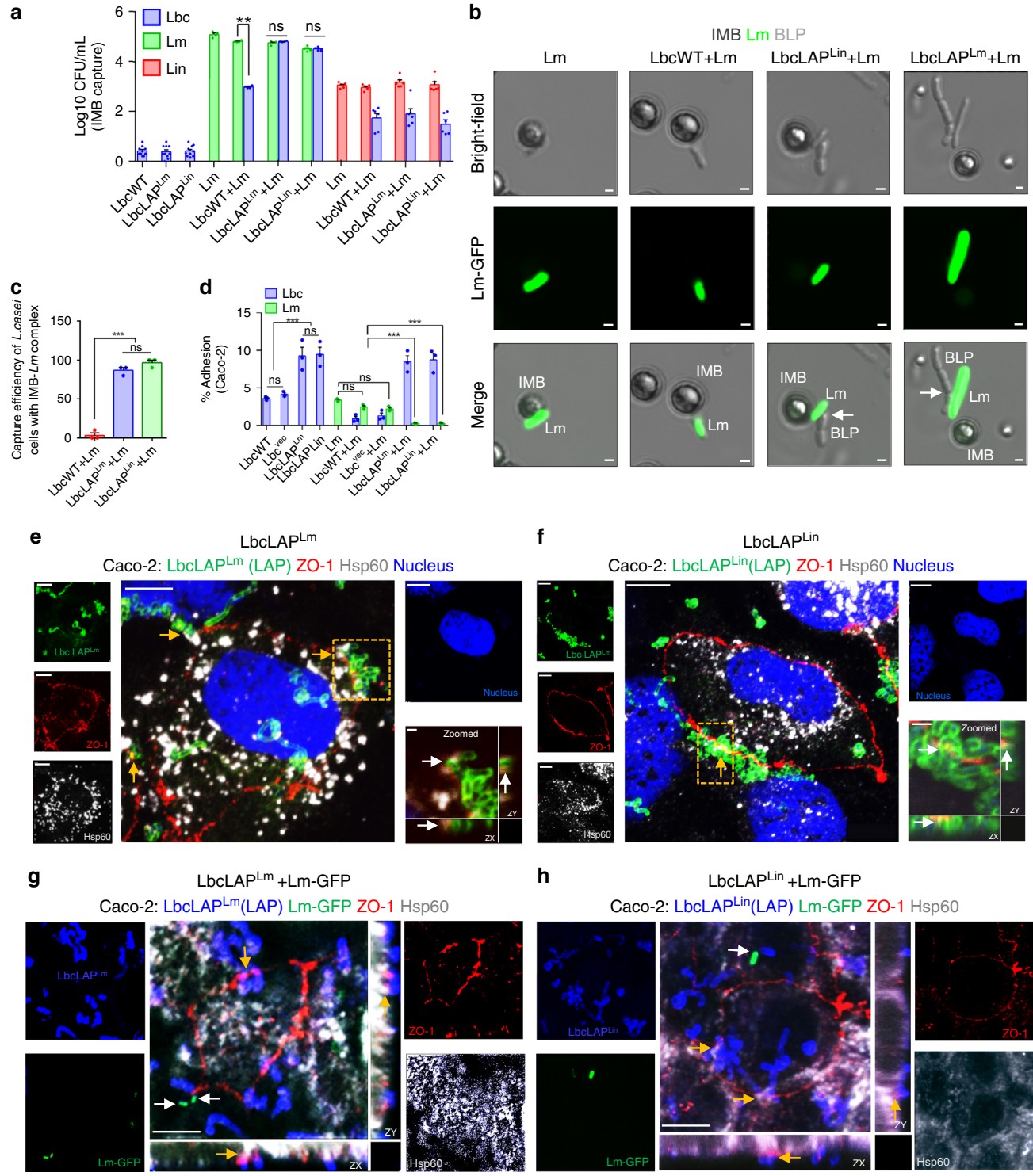

mucus-producing GCs during *Lm* infection (which may be due to the increased GCs in BLP-treated mice and also a consequence of lower *Lm* burden in the intestinal tissues), and limits epithelial proliferative and apoptotic responses thereby preventing *Lm*-induced intestinal epithelial injury.

**BLP blocks *Lm* from disrupting intestinal epithelial cell–cell junctional integrity.** We previously demonstrated that *Lm* LAP induces epithelial barrier dysfunction and promotes *Lm* translocation[31]. To better define the mechanism of protection afforded by the BLP strains, we hypothesized that BLP strains

prevent *Lm*-induced cell–cell junctional barrier defects that restrict *Lm* translocation. We first tested this hypothesis in a transwell set up of Caco-2 monolayers. Cells exposed to *Lm* or treated with LbcWT strain before *Lm* exposure showed a significant drop (17–18%) in trans-epithelial electrical resistance (TEER) (Fig. 7a). In contrast, pretreatment of cells with the BLP strains prevented ~52% of *Lm*-induced loss of TEER. Similarly, pretreatment of cells with BLP strains but not the LbcWT strain prevented ~62% of *Lm*-induced increase in epithelial permeability to the paracellular probe, FITC-dextran (4 kDa; FD4) (Fig. 7b).

**Fig. 4 BLP displays increased co-aggregation with *Lm* and competitively excludes *Lm* by occupying the epithelial Hsp60 receptor. a, b, c** Increased co-aggregation of BLP strains in co-incubated suspensions containing equal numbers of BLP+*Lm* cells (*n* = 4) captured via *Listeria*-specific immunomagnetic beads (IMB) but not with BLP+*L. innocua* (*Lin*, *n* = 6) cells. *L. casei* (alone) (*n* = 10). Micrographs (**b**) showing co-aggregated BLP cells (arrows) with IMB-captured *Lm* cells expressing GFP (**b**). Bars, 1 μm. **c** Measurements from **b**. Each point represents an average of ten fields (%) from each of the three independent experiments (*n* = 30). **d** Increased adhesion of the BLP strains (MOE 50, 1 h) exposed alone or co-incubated with *Lm* (1:1 ratio, MOI 50 for each, 1 h) with concomitantly decreased adhesion of *Lm* to Caco-2 cells (*n* = 3). **e–h** Micrographs showing co-localization of LbcLAP^Lm (**e**, yellow arrows) or LbcLAP^Lin (**f**, yellow arrows) with surface Hsp60 of mono-incubated Caco-2 cells (MOE 50, 1 h for each) immunostained for LAP (green, to stain the BLP strains), ZO-1 (red, cell periphery), Hsp60 (white, host cell receptor). Bars, 10 μm. The boxed areas are enlarged (right, bottom panel) to visualize the co-localized BLP cells and Hsp60 host cell receptors (white arrows) expressed at the peripheral ZO-1(red). Bars, 1 μm. **g, h** Micrographs showing co-localization of LbcLAP^Lm (**g**, yellow arrows) or LbcLAP^Lin (**h**, yellow arrows) with surface Hsp60 on Caco-2 cells (MOE 50, 1 h) co-incubated with *Lm*-GFP (1:1 ratio, MOI 50, 1 h) immunostained for LAP (blue, to stain the BLP strains), ZO-1 (red, cell periphery), Hsp60 (white, host cell receptor), and *Lm* (green, white arrows). Bars, 10 μm. Separated channels in **e–h** are shown individually to the left and right of the merged images. Data in **a, c, d** represent the mean ± SEM of biologically independent samples (*n*) from three independent experiments. The two-way ANOVA test followed by Tukey's multiple comparisons was used. For all analyses, ***P < 0.001; **P < 0.01; ns no significance. Images in **e–h** are representative of five different fields from three independent experiments. Source data are provided as a Source Data file.

To assess whether the BLP strains were effective in preventing *Lm*-induced intestinal barrier dysfunction in vivo, we orally administered *Lm*–infected mice with FD4, 4–5 h before sacrifice, and measured its concentration in serum and the urine[31]. Relative to the naive uninfected controls, the FD4 permeability was significantly increased by ~86% and 40% in the serum (Fig. 7c) and by 266% and 200% in the urine (Fig. 7d) in naive or LbcWT-treated mice after *Lm* challenge at 48 hpi, respectively. In contrast, the FD4 permeability was significantly lower in mice treated with the BLP strains at 48 hpi and did not increase significantly relative to the uninfected controls. Of note, the FD4 permeability in pre-*Lm*-challenged LbcWT or BLP-treated mice did not increase in the sera or urine, relative to naive controls (Fig. 7c, d). These data demonstrate that the BLP strains prevent *Lm*-induced intestinal epithelial permeability in vitro and in vivo.

The interaction of *Lm* LAP with its cognate receptor; Hsp60, results in activation of MLCK which phosphorylates MLC for cellular redistribution of cell–cell junctional proteins (claudin-1, occludin, and E-cadherin) to promote bacterial translocation[31]. However, the protection afforded by the BLP strains led us to hypothesize that the increased adhesion and intimate contact of BLP strain with the epithelial cells may block *Lm*-LAP access to its receptor (Hsp60) and in turn may diminish *Lm*-induced MLCK activation, MLC phosphorylation, and cell–cell junctional barrier defects. Consistent with this hypothesis, immunostaining of Caco-2 monolayers exposed to *Lm* or treated with LbcWT strains before *Lm* exposure, displayed significantly increased MLCK expression (3 to 4-fold increase) and MLC phosphorylation (approximately fourfold increase, Fig. 7e and Supplementary Fig. 6a). This correlated with severe mislocalization of epithelial junctional proteins as revealed by a significantly increased (three-to fourfold) number of cells containing intracellular claudin-1, occludin, and E-cadherin puncta (Fig. 7e and Supplementary Fig. 6b, c). In contrast, pretreatment of Caco-2 monolayers with BLP strains significantly prevented *Lm*-induced MLCK activation and MLC phosphorylation and these cells showed relatively undisturbed cell–cell junctional proteins, similar to uninfected controls. Additionally, in agreement with our other results (Fig. 1k–m), significantly lower numbers (~95% reduction) of adhered *Lm* were observed in Caco-2 monolayers pretreated with BLP strains compared with the *Lm* numbers in cells pretreated with LbcWT strains (Fig. 7e and Supplementary Fig. 6c, d).

Likewise, the ileal tissues of naive and LbcWT-treated mice at 48 hpi showed significantly increased MLCK and P-MLC expression (3.5 to fourfold increase) within the apical perijunctional actomyosin ring which interfaces directly with the TJ and AJ (Fig. 7f and Supplementary Fig. 6e). This correlated with severe mislocalization (endocytosis) of junctional proteins in the

ilea of these mice as revealed by a significantly increased (three-to fourfold) number of cells containing intracellular puncta of claudin-1, occludin, and E-cadherin (Fig. 7f and Supplementary Fig. 6f). In contrast, ilea of BLP-treated mice at 48 hpi showed basal levels of MLCK activation, MLC phosphorylation, and firm localization of the cell–cell junctional proteins similar to uninfected naive controls. Importantly, the treatment of Caco-2 cells or mice with LbcWT or the BLP alone did not cause epithelial MLCK activation, MLC phosphorylation, or mislocalization of cell–cell junctional proteins (Supplementary Fig. 7a–f).

Collectively, these data suggest that the intimate contact of BLP strain with the epithelial cells but not the LbcWT strain blocks *Lm* and *Lm*-induced MLCK activation, MLC phosphorylation, and preserves epithelial cell–cell junctional integrity thus further restricts *Lm* translocation across the intestinal epithelial cell barrier.

**BLP prevents *Lm*-induced NF-κB activation and modulates cytokine production to maintain intestinal immune homeostasis.** *Lm* LAP binds to Hsp60 (receptor) which activates the canonical NF-κB signaling to upregulate tumor necrosis factor-alpha (TNFα) and interleukin (IL)-6 production for increased intestinal epithelial permeability[31]. However, the decreased epithelial permeability in BLP-treated *Lm*-challenged mice led us to investigate whether the BLP prevents *Lm*-induced NF-κB activation and TNFα and IL-6 production.

Examination of the nuclear abundance of NF-κB (p65) and P-p65 by immunostaining showed significantly increased nuclear translocation of p65 (15–20-fold) and phosphorylated-p65 (P-p65, 12–15-fold); a hallmark of NF-κB activation, in Caco-2 cells exposed to *Lm* or treated with LbcWT strain before *Lm* exposure (Fig, 8a and Supplementary Fig. 8a–c). In contrast, pretreatment of Caco-2 cells with the BLP strains prevented *Lm*-induced nuclear translocation of p65 and P-p65. Similarly, a significantly increased nuclear abundance of p65 (six- to sevenfold, Fig. 8b, c) and P-p65 (10–15-fold, Fig. 8d, e) was observed in the intestinal epithelial cells (IECs) of the ilea of naive and LbcWT-treated mice at 48 hpi. However, only a few nuclear-positive p65 and P-p65 IECs were found and most of the p65 were sequestered in the cytoplasm of IECs of BLP-treated mice at 48 hpi. Consistent with these observations, the levels of TNFα (Fig. 8f) and IL-6 (Fig. 8g) were markedly reduced in the ileal tissues of BLP-treated mice, relative to naive and LbcWT-treated mice at 48 hpi. The ilea of pre-*Lm*-challenged LbcWT or BLP-treated mice showed only a few nuclear-positive p65 and P-p65 IECs similar to naive mice (Supplementary Fig. 8d–g) and did not significantly change the levels of TNFα (Fig. 8f) and IL-6 (Fig. 8g), relative to naive mice.

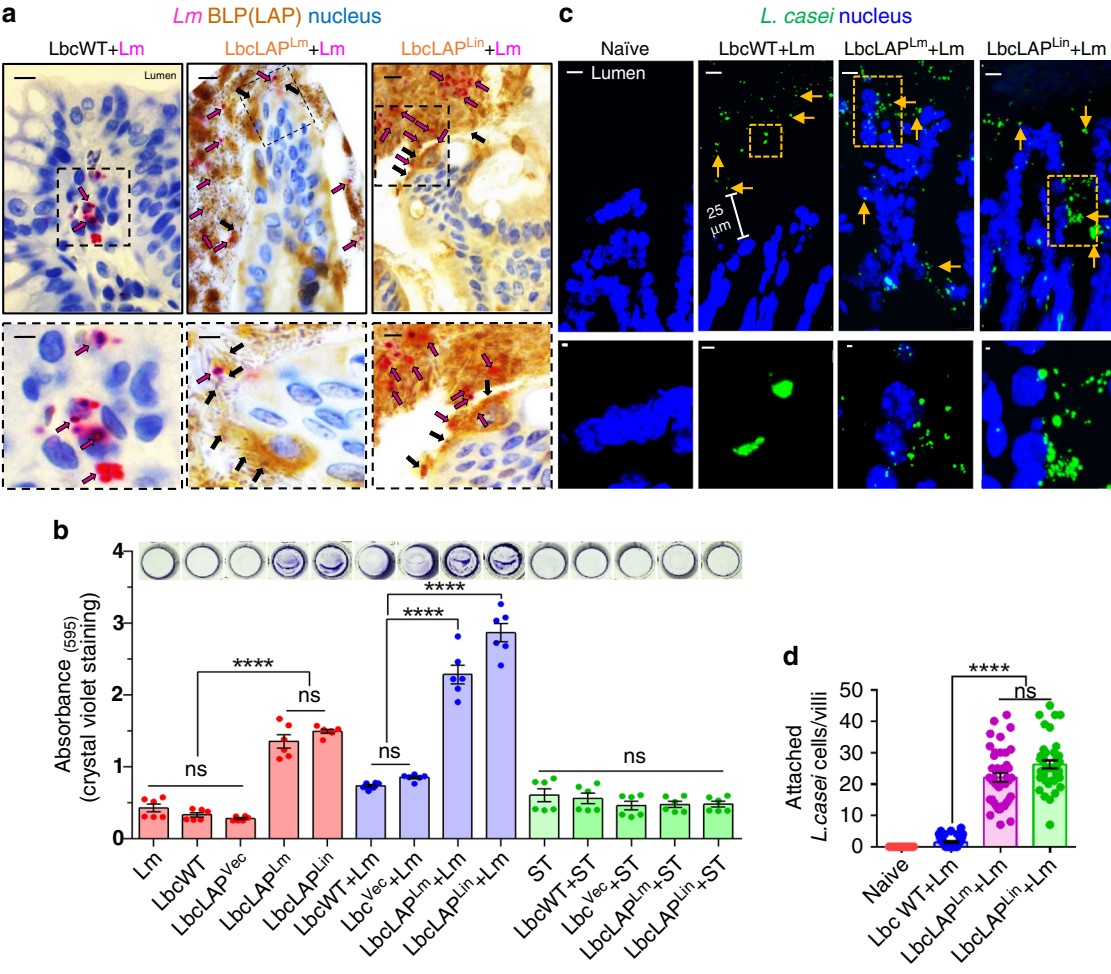

**Fig. 5 BLP forms increased biofilm and restrict *Lm* cells at the lumen and epithelial surface. a** Micrographs of colonic villi of LbcWT or BLP-treated (10 days) mice at 48 hpi dual immunostained for *Listeria* (anti-*Lm* pAb, pink rods, pink arrows) and LAP (anti-LAP mAb to stain the BLP strains, brown, black arrows) and counterstained with hematoxylin to stain the nucleus (blue). Bars, 10 µm. The boxed areas are enlarged (bottom panel). Bars, 1 µm. Translocated *Lm* is observed in the lamina propria (pink arrows, left panel) in LbcWT-treated mice but confined in the lumen (pink arrows, middle and right panels) in BLP-treated mice. **b** Increased biofilm formation (Abs 595 nm, mean ± SEM) of BLP strains as measured by crystal violet staining in monoculture and co-culture with *Lm*, but not in co-culture with *Salmonella enterica* serovar Typhimurium (ST) grown in microtiter plates. Images (top panel) show crystal violet stained biofilms of representative wells for each treatment. Data represent three independent experiments obtained from *n* = 6 independent microtiter plate wells. **c**, Micrographs of colonic villi of LbcWT or BLP-treated mice (10 days) at 48 hpi after fluorescent in-situ hybridization (FISH, **c**) using *L. casei*-specific 16 s rDNA probe (green) and counterstained to visualize the nucleus (DAPI). In LbcWT-treated mice, bacteria were seen in the mucus layer (arrows) in the lumen while BLP-treated mice bacterial clusters (arrows) are primarily on the surface of epithelial cells. Bars, 10 µm. The enlargements of the boxed areas in each image are shown in the bottom panel. Bars, 1 µm. **d** Quantitative measurements (mean ± SEM) of attached *L. casei* cells to the epithelial cells from FISH images (**c**). Each point represents an individual villus, *n* = 40 villi from four mice for each treatment. For panels **b** and **d**, the one-way ANOVA test followed by Tukey's multiple comparisons was used For all analyses, ****$P < 0.0001$. Images are representative of ten different fields (**a**) or villi (**c**) from four independent mice for each treatment. Source data are provided as a Source Data file.

These results suggest that the BLP but not the LbcWT stains prevent the *Lm*-induced epithelial NF-κB activation and production of TNFα and IL-6, consistent with the decreased epithelial permeability in these mice (Fig. 7c, d).

Early production of interferon-gamma (IFNγ) is a critical step for generating an immune response and controlling *Lm* infection[50]. In the intestine, IFNγ is also involved in tissue homeostasis[51] and the transcription factor STAT4 promotes *Lm*-induced IFNγ production[52]. Therefore, we assessed the levels of IFNγ in BLP-treated mice. Relative to naive or LbcWT-treated mice, the levels of IFNγ were significantly increased (100-200%) in the ilea of BLP-treated mice pre- or 48 h post-*Lm* challenge (Fig. 8h).

*Lactobacillus* species have been shown to induce pathogen clearance and reduce intestinal inflammation by enhanced production of anti-inflammatory cytokines such as IL-10 and transforming growth factor-beta (TGFβ)[53,54]. Thus, we next examined the production of IL-10 and TGFβ in BLP-treated mice. Strikingly, the naive and LbcWT-treated mice down-regulated the proportion of IL-10 (Fig. 8i and Supplementary Fig. 8h) and TGFβ (Fig. 8j and Supplementary Fig. 8i)-positive cells in the ilea at 48 hpi. In contrast, we observed a significant increase in the proportion of IL-10 (~150%) and TGFβ (~75%)-positive cells in the ilea of BLP-treated mice pre- or 48 h post Lm challenge, consistent with reduced histopathology and inflammation in these mice (Fig. 5a). Taken together, these results suggest that oral administration of BLP promotes the production of IFNγ for effective *Lm* clearance and upregulates IL-10 and TGFβ that prevent excessive inflammation, thus maintain intestinal immune homeostasis.

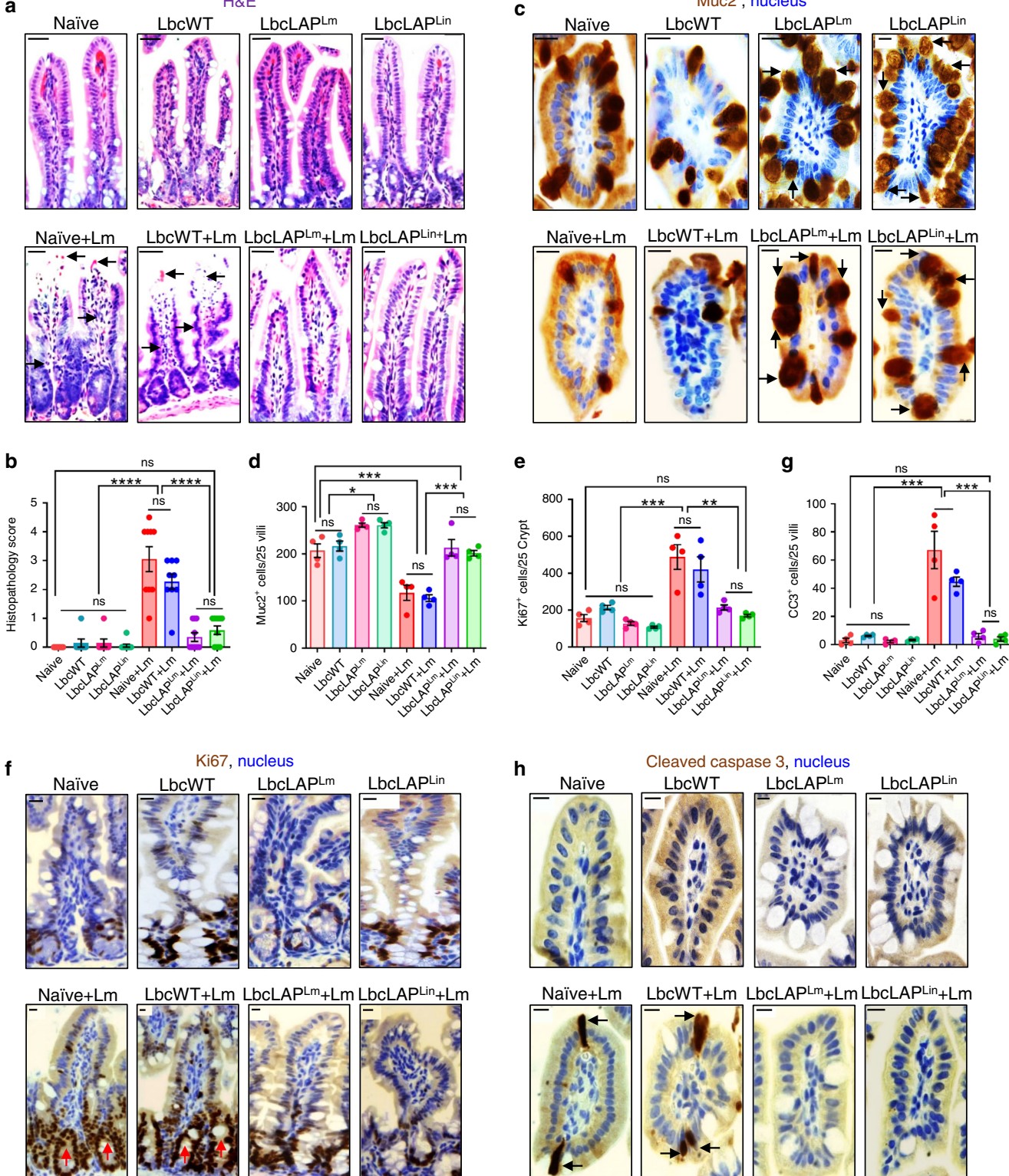

**BLP promotes immunomodulation to protect the host from *Lm* infection.** Next, we assessed whether the BLP strains modulate the intestinal mucosal infiltration of the immune cells in the underlying lamina propria by immunostaining. Relative to naive uninfected mice, we observed a significant increase in the numbers of infiltrated F4/80$^+$ macrophages (~225%, Supplementary Fig. 9a), CD3$^+$ T cells (~120%, Supplementary Fig. 9b) and CD8$^+$ T cells (~2000%, Supplementary Fig. 9c) in the lamina propria of naive or LbcWT-treated mice at 48 h post *Lm* challenge, in line

with the increased *Lm* burdens (Fig. 2f–j), inflammation and histopathology scores (Fig. 5a, b). In contrast, relative to naive mice, the population of F4/80$^+$ macrophages did not change significantly in BLP-treated mice pre or post *Lm* challenge (Supplementary Fig. 9a). Increased numbers of CD3$^+$ T cells (~80%) were observed in BLP-treated mice pre-Lm challenge and were maintained in these mice post *Lm* challenge, relative to naive mice (Supplementary Fig. 9b). Subset analyses of T cells showed fewer CD8$^+$ T cells in BLP-treated mice 48 h post *Lm*

**Fig. 6 BLP prevents *Lm* from causing intestinal barrier loss by maintaining mucus-producing goblet cells and limiting epithelial apoptotic and proliferative cells. a, b** Representative H&E-stained micrographs (bars, 25 μm) (**a**) and the histological score (**b**, each point represents an individual mouse) of ileal tissue sections from control (mock-treated) uninfected naive mice or *L. casei*-treated (10 days, LbcWT or BLP) pre- or post-*Lm* challenge at 48 hpi (*n* = 9, 7, 7, 9, 9, 9, 10, 11 mice for each group, respectively). Arrows point to the loss of villous epithelial cells and increased polymorphonuclear and mononuclear cells infiltrating the base of the villous lamina propria in naive (naive + *Lm*) and LbcWT-treated mice (LbcWT + *Lm*) at 48 hpi. **c–h** Representative immunohistochemical micrographs of the ileum stained for Muc2 (**c**, brown), Ki67 (**f**, brown) and cleaved caspase-3 (**h**, brown), nuclei (blue) from control (mock-treated) uninfected naive mice or *L. casei*-treated (10 days, LbcWT or BLP) pre or post-*Lm* challenge at 48 hpi. Bars, 10 μm. Quantification of Muc2 (**d**), Ki67 (**e**), and CC3 (**g**)-positive cells, each point represents an individual mouse, four mice per group, *n* = 100 villi. Arrows point to increased numbers of Muc2 (**c**) in BLP-treated mice (pre- or post-*Lm* challenge), and increased numbers of Ki67 (**f**) and CC3-positive cells (**h**) in naive or LbcWT-treated mice at 48 hpi. Data in **b, d, e**, and **g** represent the mean ± SEM and statistical significance was determined by using the one-way ANOVA test followed by Tukey's multiple comparisons. For all analyses, ****$P < 0.0001$; ***$P < 0.001$; **$P < 0.01$; *$P < 0.05$; ns no significance. Source data are provided as a Source Data file.

challenge (Supplementary Fig. 9c) while significantly increased numbers of CD4[+] T cells (~66%, Fig. 9a, b) and CD4[+] FOXP3[+] regulatory T cells (~150%, Fig. 9c, d), relative to naive or LbcWT-treated mice pre- or 48 h post *Lm* challenge.

*Lactobacillus* strains have been shown to induce CD11c[+] dendritic cell population which in turn promotes the generation of CD4[+]FOXP3[+] T cells[55]. Thus, we analyzed the levels of CD11c[+] dendritic cells, which markedly increased (~300%) in the lamina propria of BLP-treated mice compared to the naive or LbcWT-treated pre- or 48 h post *Lm* challenge (Fig. 9e, f). Probiotic bacteria activate NK cells to induce an adaptive immune response against pathogens[56]. The NK cells are also a major producer of IFNγ, a cytokine that is required for effective bacterial clearance[48]. Monitoring of NK cell population showed significantly increased (~100%) NKp46[+] cells in the lamina propria of BLP-treated mice pre- or 48 h post *Lm* challenge, relative to the naive or LbcWT-treated mice (Fig. 9g, h), consistent with increased IFNγ production (Fig. 8h) in these mice. Cumulatively, these data suggest that BLP promotes intestinal immunoregulatory functions by enhancing FOXP3[+] T-regulatory cells and CD11c[+] dendritic cells and immunostimulatory functions by recruiting NK cells for effective clearance of *Lm*.

## Discussion

A vast majority (99%) of listeriosis cases are due to the consumption of contaminated food; therefore, restricting *Lm* at the gastrointestinal phase of infection is the best approach to limit the spread of the pathogen to the deeper tissues and consequent lethality.

The use of probiotic bacteria has been proposed as a rational approach to counteract intestinal pathogens[8,16,57]. A previous study has shown that a bacteriocin produced by *Lactobacillus salivarious* was able to control *Lm* infection in mice[47]. However, other major mechanisms of the proposed action of probiotic bacteria such as increased adhesion to the intestinal mucosa and concomitant inhibition of pathogen adhesion, competitive exclusion, prevention of pathogen-induced disruption of epithelial integrity, and modulation of the immune system have thus far not been directly shown to confer resistance to *Lm*. This is mainly because probiotic bacteria have limited success to prevent *Lm* infection due to species or strain-specific activity of probiotics, and their inadequate epithelial colonization[7,15,38,58,59].

In this study, we took a molecularly targeted approach and rationally designed the BLP strains that express LAP to prevent *Lm* infection by taking advantage of ligand-receptor interactions. Due to a very high binding affinity of both *Lm* LAP and *Lin* LAP (ligand) to the receptor, Hsp60 ($1.68 \times 10^{-8}$ M, and $3.12 \times 10^{-8}$ M, respectively)[36,60,61], we chose to use LAP as the bioengineered ligand. We bioengineered two probiotic strains, one that expresses the LAP from *Lin* and the other from *Lm* on their surface through anchoring to the PrtP on the cell wall[43]. Our data

suggest that the expression of LAP in the engineered strains not only aids the intimate contact of the BLP strains with the intestinal epithelial cells for promoting enhanced and prolonged probiotic colonization but also excludes the interaction of *Lm* with the host cells. We demonstrated that BLP strains but not the parental LbcWT strain dramatically reduce *Lm* adhesion, invasion, and translocation in vitro and in vivo to mitigate lethal *Lm* infection in an established A/J mouse model that is highly sensitive to *Lm* infection[31,37,46,47]. This approach of engineering a probiotic strain with an adhesion protein from a nonpathogenic bacterium to exclude a pathogen significantly enhances the prophylactic use of such bioengineered probiotic bacteria without raising serious health or regulatory concerns and thus their potential as preventive agents against listeriosis.

The observed robust antilisterial effect of our engineered probiotics is based on three plausible mechanisms (Fig. 10): (i) Competitive exclusion, (ii) improved intestinal barrier function, and (iii) contact-dependent immunomodulation.

It is proposed that probiotics may compete with pathogens for adhesion sites[8], however, to our knowledge[8], there is no direct evidence of this phenomenon. Our data demonstrate that the BLP, but not the LbcWT strains were able to co-aggregate with *Lm* and occupy the membrane expressed epithelial Hsp60 receptor sites on epithelial cells to competitively exclude *Lm*. Additionally, our observations suggest that the BLP strains were able to pass through the layers of gut microbiota and the loosely and tightly adherent mucus to interact directly with epithelial cells. Thus, in contrast to parental LbcWT, the close contact and proximity of the BLP strain to the intestinal epithelial cells increase the opportunity for interacting with the host resulting in better executions of contact-dependent mechanisms (competitive exclusion and immunomodulation) to exert their intended beneficial effects[62]. This study thus provides direct evidence that rational engineering of probiotic strains allows them to outcompete and diminish the colonization of the pathogens by competing for the receptor-binding adhesion sites.

The mucus produced by GC's serves as an important innate defense. Many intestinal pathogens have evolved mechanisms that can circumvent the mucus protection to reach the epithelium[63]. Consistent with a previous report[48], we observed that *Lm* infection leads to depletion of GC's and increases epithelial proliferation. This may restrict the luminally accessible E-cadherin sites at the mucus-secreting GC's which serves as a receptor for *Lm*-InlA[33]. Although the depletion of GC's during *Lm* infection may provide a temporary benefit to the host by blocking the access of *Lm* to its host receptor; this also leads to a reduction of the protective mucus barrier. Interestingly, we found that treatment with the BLP strains but not the LbcWT strains promoted GC and MUC2[+] GC counts thus strengthening the

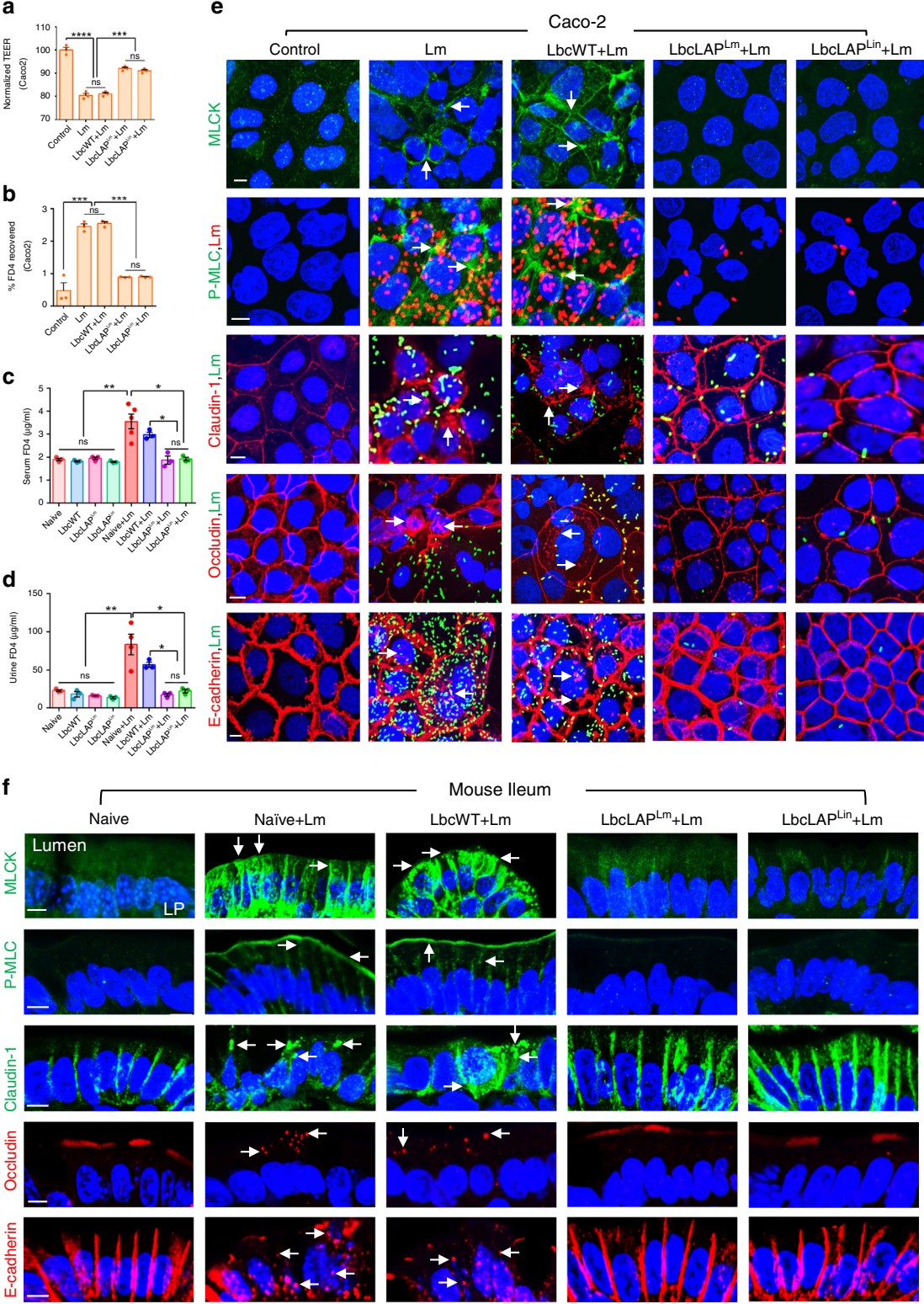

mucus barrier and limited *Lm*-induced loss of MUC2⁺ GC's and epithelial proliferative and apoptotic responses.

The intestinal epithelial cells serve as the first line of defense and prevent the unrestricted passage of bacteria[64]. Both in vitro and in vivo, treatment with the BLP but not the LbcWT strains prevented *Lm*-induced intestinal permeability, NF-κB, and MLCK activation and subsequent phosphorylation of perijunctional MLC. This functional preservation directly correlated with

BLP-mediated inhibition of *Lm*-induced redistribution of TJ and AJ pools and maintenance of the cell junctional architecture of claudin-1, occludin, and E-cadherin. Our data further imply that the intimate association of BLP strains despite *Lm* infection prevents *Lm* from gaining the physical proximity required for activation of the downstream signaling pathways to breach the epithelial barrier. The preservation of MUC2⁺GC's and barrier functions by the BLP strains may be a consequential response due

**Fig. 7 BLP blocks *Lm* from causing disturbance of intestinal epithelial cell–cell junctional integrity. a, b** TEER of Caco-2 cell monolayer treated with LbcWT or BLP strains (MOE; 10, 24 h) before *Lm* exposure (MOI; 50, 2 h) (**a**) and on the apical (AP)-to-basolateral (BL) flux of FD4 permeability (**b**). Data in **a** and **b** represent mean ± SEM from *n* = 3 biologically independent samples. **c, d** FD4 gut permeability of control (mock-treated) naive uninfected mice or *L. casei*-treated (10 days, LbcWT or BLP) pre- or post-*Lm* challenge (48 hpi) in serum (**c**) and urine (**d**). Each point represents an individual mouse. Data (**c, d**) represent mean ± SEM of *n* = 3 mice for all groups except *Lm* group, *n* = 5 mice. **e** Immunofluorescence micrographs of Caco-2 cells showing increased expression of MLCK and P-MLC (green; arrows) and mislocalization (intracellular puncta) of claudin-1, occludin, and E-cadherin (red; arrows) in cells exposed with *Lm* (MOI; 50, 2 h) or treated with LbcWT (MOE; 10, 24 h) before *Lm* exposure but baseline expression of MLCK and P-MLC and intact localization of occludin, claudin-1, and E-cadherin in cells treated to BLP strains before *Lm* exposure, relative to untreated (control) cells. Nuclei; DAPI, blue. *Lm* cells are double immunostained in red in the P-MLC panel and green in occludin, claudin-1, and E-cadherin panels. Images are representative of five different fields. Bars, 10 μm. **f** Immunofluorescence micrographs of the ileal tissues showing increased expression of MLCK and P-MLC (green; arrows) and mislocalization (intracellular puncta) of claudin-1 (green; arrows), occludin and E-cadherin (red; arrows) in naive or LbcWT-treated (10 days) but baseline expression of MLCK and P-MLC and intact localization of occludin, claudin-1, and E-cadherin in BLP-treated mice (10 days) at 48 hpi, relative to uninfected naive mice. Nuclei; DAPI, blue. Images are representative of five different fields from *n* = 3 mice per treatment. Bars, 10 μm. LP lamina propria. Data in **a–e** are from three independent experiments. For all analysis, the one-way ANOVA test followed by a Tukey's multiple comparisons was used; ****$P < 0.0001$; ***$P < 0.001$; **$P < 0.01$; *$P < 0.05$; ns no significance. Source data are provided as a Source Data file.

to lower burdens of *Lm* in the intestine. However, to our knowledge[34], a dose–response comparing GC counts and junctional barrier function with an increasing infectious dose of *Lm* has not been examined previously. Nevertheless, these data imply that the protection afforded by the BLP extends to preventing multiple aspects of *Lm*-induced intestinal epithelial insult.

One remaining question is why the interaction of LAP expressed on the BLP strains with the host Hsp60 does not lead to increased intestinal permeability? One plausible explanation is that the *Lm* LAP or the *Lin* LAP was expressed on the BLP strains through anchoring to the PrtP on the cell wall[43]. In contrast, in *Lm*, the secreted LAP spontaneously re-associates with the cell wall[36,37] with a yet unknown receptor, which is currently under investigation. The direct anchoring of the LAP to the PrtP on the cell wall of the BLP strains may affect the protein folding (exposed amino acid residues) and its display on the cell surface which may be different from that of the cell surface of *Lm*. This differential protein folding or spatial structural display may affect receptor (Hsp60) interaction and consequent downstream signaling events. Alternatively, increased adhesion and intimate epithelial contact by the BLP strain may be sufficient to promote the inherent properties of the probiotic bacteria to maintain intestinal barrier integrity that masks or supersedes the adverse effect of LAP expressed on the BLP strains.

The intestinal epithelium also actively participates in immune reactions and several lines of evidence suggest that probiotic bacteria exert immunomodulatory effects[12,40,65] and promote gut health[13]. The BLP significantly increased the proportion of intestinal IL-10 and TGFβ-producing cells. IL-10 preserves the intestinal mucus barrier by suppressing protein misfolding and endoplasmic reticulum stress in GCs[66]. Thus, the treatment with BLP may also promote epithelial barrier integrity by preventing GC stress, and suppressing excessive inflammatory responses in the intestine. Additionally, the BLP- significantly increased NK cells and IFNγ levels that are required for effective clearance of *Lm* and were also able to activate immunoregulatory cytokines (IL-10 and TGFβ) to thwart *Lm*-mediated overt inflammatory response and the resulting intestinal inflammatory and pathological damages. BLP treatment also increased the intestinal CD4+FOXP3+ Treg cells and CD11c+ dendritic that helps to maintain epithelial immune homeostasis[55,67]. These data suggest that rational engineering of probiotics significantly enhances their immunomodulatory properties and that the BLP strains may be useful in a variety of gastrointestinal and systemic diseases, including but not limited to inflammatory bowel disease (IBD), graft-versus-host disease, and coeliac disease.

The major limitations of probiotics for prophylactic or therapeutic use are their poor ability to colonize the intestine[15]. Our

BLP displays significantly improved colonization and persistence in vivo. These data have important implications and suggest that the rationally designed BLP strain using the receptor–ligand bioengineering strategy can be useful in enhancing the engraftment of probiotics in humans. The BLP strain can colonize the host for an extended period to provide their intended beneficial effects, thus daily administration may not be necessary. Our data suggest that once a week administration would be sufficient to achieve protection against *Lm* infection. However, these engineered probiotics may colonize the host for extended periods, permitting the genetically modified microorganism to be released into the external environment. Therefore, suitable biological containment systems such as the creation of an auxotroph for an essential metabolic gene can prevent BLP survival outside the host thus increase the utility of our approach[18,21].

In summary, we used a molecularly targeted approach to create a next-generation bioengineered *L. casei* strain with an adhesion protein from a nonpathogenic *Listeria* to prevent *Lm* infection through competitive exclusion, maintenance of intestinal epithelial barrier functions and contact-dependent immunomodulation. This represents a new paradigm to paving the translational path forward for the preventive application of this engineered probiotic with favorable regulatory compliance. Beyond the application of the BLP strains to prevent listeriosis in high-risk immunosuppressed populations, this receptor–ligand bioengineering strategy provides a pathogen-specific targeted approach to enhance the specificity of probiotic action and extend the health beneficial effects inherent to probiotic lactobacilli.

## Methods

**Bacterial strains, plasmids, and growth conditions.** Bacterial strains and plasmids used in this study are listed in Supplementary Table 1. All *Listeria* species were grown in tryptic soy broth containing 0.6% yeast extract (TSBYE; Becton Dickinson) or in Luria-Bertani broth (LB, 0.5% NaCl, 1% tryptone peptone, and 0.5% yeast extract) at 37 °C with shaking for 16–18 h. The *L. monocytogenes* (*Lm*) F4244 (WT, serovar 4b) and *L. innocua* (*Lin*) F4248 were grown in TSBYE. The isogenic *lap*-deficient insertional mutant strain (*lap⁻*, KB208) was grown in TSBYE with erythromycin (5 μg/mL) at 42 °C. The *lap⁻* stain complemented with the *Lm lap* (*lap⁻*+*lap^Lm*, CKB208) was grown in TSBYE containing erythromycin (5 μg/ml) and chloramphenicol (Cm; 5 μg/ml) at 37 °C. The *Lm* F4244 (WT) strain expressing the green fluorescent protein (GFP) were grown in LB containing erythromycin (5 μg/ml) at 37 °C. *S. enterica* serovar Typhimurium ver. Copenhagen was grown in TSBYE at 37 °C.

To express the *Lin* LAP in the isogenic *lap*-deficient insertional mutant strain (*lap⁻*, KB208), the *Lin lap* gene was cloned in the *Listeria* expression vector pMGS101[68] and electrotransformed into the *lap*-deficient insertional mutant strain (*lap⁻*, KB208) and the resulting strain was designated *lap⁻*+*lap^Lin* and was grown in TSBYE containing erythromycin (5 μg/ml) and chloramphenicol (Cm; 5 μg/ml) at 37 °C.

The *Lactobacillus casei* ATCC334 wild-type (LbcWT) strain was used as a host to express LAP from *Lin* and *Lm* were cultured in deMan Rogosa Sharpe broth (MRS, Becton Dickinson) at 37 °C for 18–20 h under anaerobic conditions. To

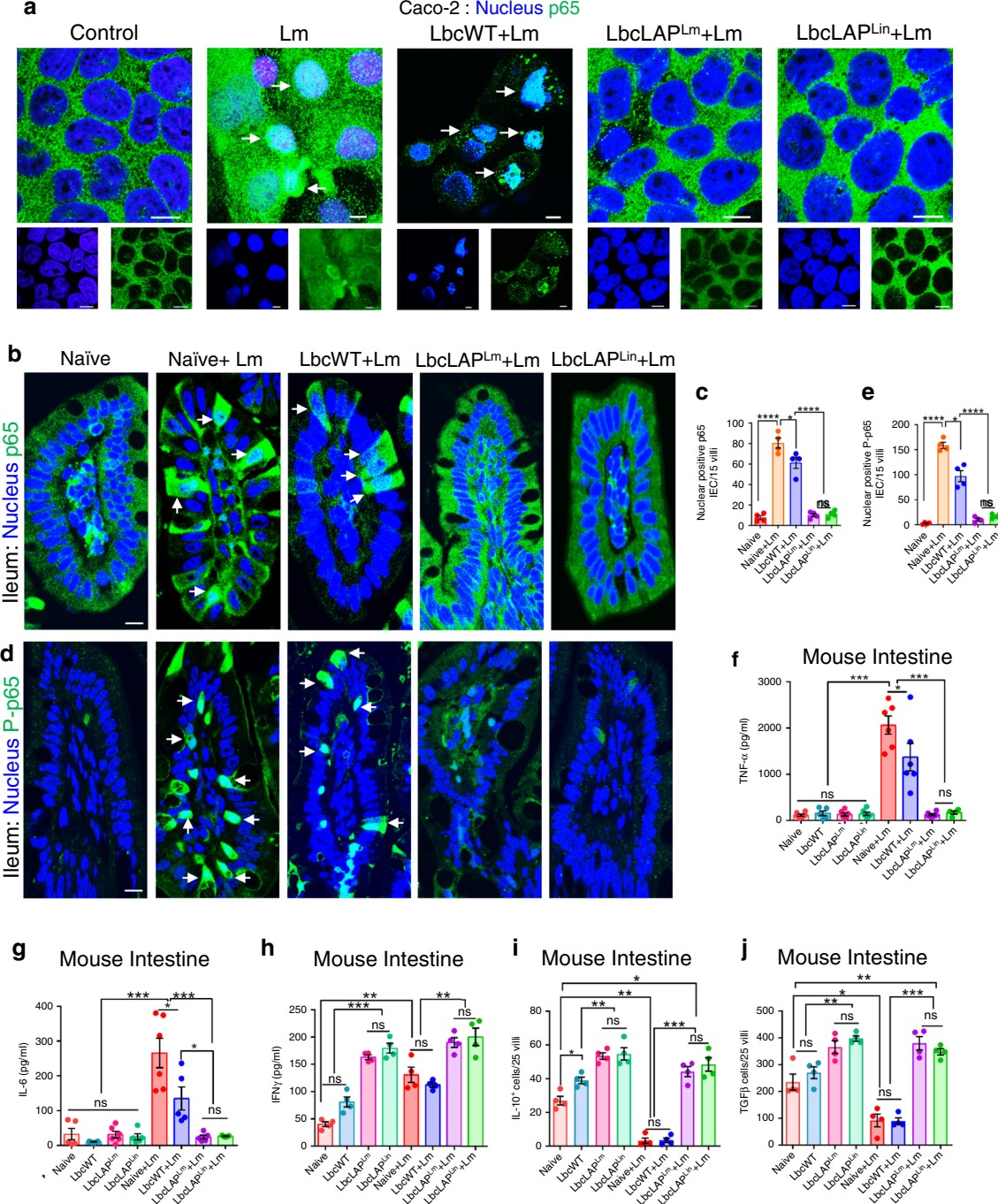

**Fig. 8 BLP prevents *Lm*-induced NF-κB activation and modulates cytokine production and immune cells to maintain intestinal immune homeostasis.** **a** Immunofluorescence micrographs showing decreased nuclear localization of p65 (**a**, green) in Caco-2 cells treated with BLP strains (MOE 10, 24 h) before *Lm* exposure (MOI 50, 1 h). Arrows indicate the nuclear localization of p65. Separated channels; bottom panels. Bars, 10 μm. Images represent five different fields from three independent experiments. **b–e** Immunofluorescence micrographs of the ileal tissues showing decreased nuclear localization of p65 (**b**, green) and P-p65 (**d**, green) in BLP-treated mice (10 days) at 48 hpi. Nuclei; DAPI, blue. Arrows indicate the nuclear localization of p65 (**b**) and P-p65 (**d**) in IEC of naive or LbcWT-treated (10 days) mice at 48 hpi. Right panels (**c**, **e**) show the quantified results (mean ± SEM) of p65 (**c**) and P-p65 (**d**) nuclear-positive IEC. Each point represents an average of 15 villi from a single mouse, four mice per group, n = 60 villi. Bars, 10 μm. **f**, **g**, **h** ELISA showing decreased TNFα (**f**, n = 6 mice for all groups, except LbcWT and LbcLAP^Lin + Lm group; n = 5 and 4 mice, respectively), and IL-6 (**g**, n = 5 mice for all groups except LbcWT, LbcLAP^Lm and naive +Lm groups; n = 6 mice) levels (mean ± SEM) in the ileal tissues of BLP-treated (10 days) mice, relative to naive or LbcWT-treated (10 days) at 48 hpi and increased IFNγ (**h**, n = 4) levels (mean ± SEM) pre- or post-*Lm* challenge at 48 hpi. Each point; individual mouse. **i**, **j** Graphs showing increased IL-10⁺ (**i**) and TGFβ⁺ (**j**) cells quantified (mean ± SEM) from immunostained ileal tissues (Supplementary Fig. 8h, i, respectively) of BLP-treated (10 days) mice pre- or post-*Lm* challenge at 48 hpi. Each point represents an average of 25 villi from a single mouse, four mice per group, n = 100 villi. Statistical significance (**c**, **e**, **f–j**) was determined by using the one-way ANOVA followed by a Tukey's multiple comparisons. For all analysis, ****P < 0.0001; ***P < 0.001; **P < 0.01; *P < 0.05; ns no significance. Source data are provided as a Source Data file.

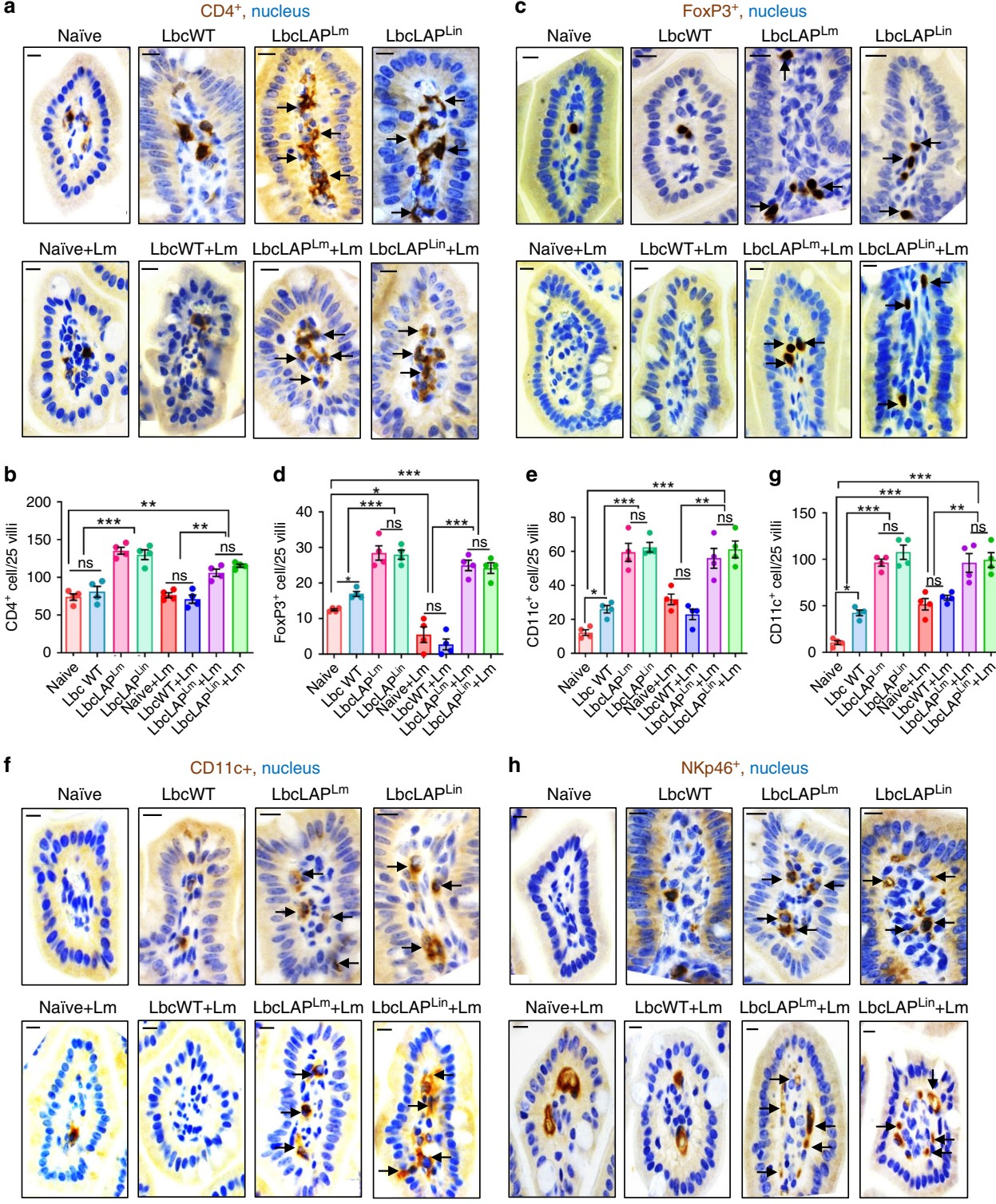

**Fig. 9 BLP modulates immune cell populations to maintain intestinal immune homeostasis. a–h** Representative immunohistochemical micrographs of ileal tissues showing increased CD4+ cells (**a**, brown, arrows), FOXP3+ T-regulatory cells (**c**, brown, arrows), CD11c+ dendritic cells (**f**, brown, arrows), and NKp46+ cells (**h**, brown, arrows) in BLP-treated mice (10 days) mice pre- or post *Lm* challenge at 48 hpi, relative to naive or LbcWT-treated (10 days). Bars, 10 μm. Quantification of CD4+ cells (**b**), FOXP3+ T-regulatory cells (**d**), CD11c+ dendritic cells (**e**), and NKp46+ cells (**g**). Each point (**b**, **d**, **e**, and **g**) represents an average (mean ± SEM) of 25 villi from a single mouse, four mice per group, $n = 100$ villi. Statistical significance (**b**, **d**, **e**, and **g**) was determined by using the one-way ANOVA followed by a Tukey's multiple comparisons. For all analysis, ***$P < 0.001$; **$P < 0.01$; *$P < 0.05$; ns no significance. Source data are provided as a Source Data file.

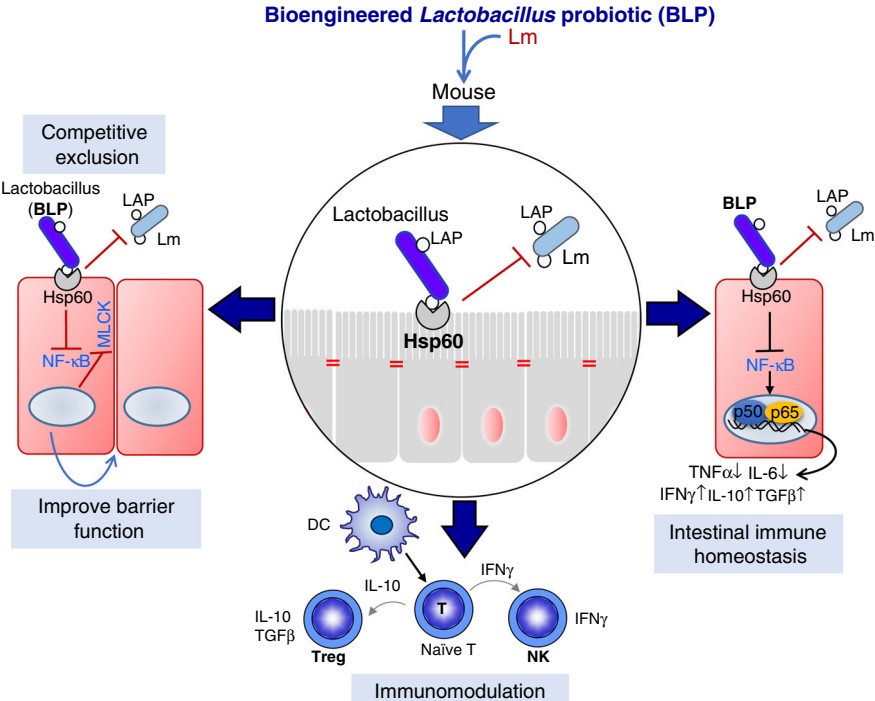

**Fig. 10 Schematics showing the mechanism of BLP-mediated protection against listeriosis.** The BLP prevents *Lm* Infection by three mechanisms (i) competitive exclusion, (ii) improved intestinal barrier function, and (iii) contact-dependent immunomodulation.

recover this strain from fecal and intestinal samples during the animal study, a vancomycin-resistant strain of *L. casei* was selected by sequentially culturing the bacterium in increasing concentrations of vancomycin (300 μg/ml). To generate the bioengineered lactobacilli expressing LAP from *Lin* and *Lm*, the entire *lap* gene (2.6 kb) from *Lm* F4244 (WT) was amplified by PCR and inserted into pLP401T[69] containing the pAmy promoter and electrotransformed[38] into the selected vancomycin-resistant *L. casei* strain.

Briefly, the genomic DNA of *Lm* F4244 and *Lin* F4248 was extracted and the entire *lap* gene (2.6 kb) from of *Lm* (*lap^Lm*) and *Lin* (*lap^Lin*) amplified with PCR using the primers: LAPN-F 5′- GACCATGGATGGCAATTAAAGAAAATG-3′ and LAPX-R-5′-GACTCGAGTCAAACACCTTTGTAAG-3′ (Integrated DNA Technologies, Supplementary Table 1)[38]. The amplified DNA products were cloned into pGEM-T Easy Vector (Promega) and designated pGEM-LAP^Lm and pGEM-LAP^Lin, respectively. The *Lactobacilli* expression vector, pLP401T containing the pAmy promoter was used[69]. The plasmids were digested with NcoI and XhoI, inserted into expression vector pLP401T, and designated pLP401T-LAP^Lm and pLP401T-LAP^Lin. To remove the terminator, which stabilizes the plasmid in *E. coli*, pLP401T-LAP^Lm/Lin was digested with NotI, and pLP401T-LAP^Lm and pLP401T-LAP^Lin were obtained via self-ligation. Self-ligated pLP401T-LAP^Lm and pLP401T-LAP^Lin were used for electroporation into competent vancomycin-resistant *L. casei* cells. Competent vancomycin-resistant *L. casei* cells were prepared by incubation of 2% culture in fresh MRS broth containing 0.5% sucrose and 0.5% glycine at 37 °C until OD600 reached to 0.5–0.8. The cells were harvested (3900×*g* for 5 min at 4 °C), washed twice with washing buffer (0.5 M sucrose, 10% glycerol), and collected. The cells were resuspended in the same washing buffer and stored at −80 °C. For electroporation, 50 μl of competent cells mixed with 0.5 μg of purified plasmid DNA in an ice-cold cuvette with a 2-mm electrode gap. The electric pulse was delivered by the Gene Pulser Xcell™ electroporation system (Bio-Rad) using the following parameter settings: 1.5 kV, 200 Ω, and 25 mF. After electroporation, competent cells were recovered in 1 ml of MRS containing 0.5 M sucrose, 20 mM MgCl₂, 2 mM CaCl₂ at 37 °C for 2 h in a water bath. Electroporated *L. casei* cells were then incubated at 37 °C for 3 h. Transformants harboring pLP401T-LAP^Lm and pLP401T-LAP^Lin were subsequently selected on MRS agar containing 2 μg/ml erythromycin that were incubated at 37 °C overnight for 72 h. The resulting bioengineered lactobacilli probiotic (BLP) expressing LAP from *Lm* and *Lin* were designated LbcLAP^Lm (AKB906) and LbcLAP^Lin (AKB907), respectively. Confirmation of the identity of the *lap* gene in the LbcLAP^Lm and LbcLAP^Lin strain was done using PCR and sequencing. The selected vancomycin-resistant *L. casei* strain carrying the pLP401T empty vector (Lbc^Vec) was used as a control. The bioengineered strains and the Lbc^Vec were maintained in MRS broth containing erythromycin (2 μg/ml) under anaerobic conditions at 37 °C.

To induce the expression of LAP, the BLP strains, were grown in modified MRS broth (1% w/v protease peptone, 0.5% w/v yeast extract, 0.2% w/v meat extract, 0.1% v/v Tween 80, 37 mM C₂H₃NaO₂, 0.8 mM MgSO₄, 0.24 mM MnSO₄, 8.8 mM

C₆H₁₄N₂O₇ in 0.2 M potassium phosphate (dibasic), pH 7.0) supplemented with mannitol (1% w/v) at 37 °C under anaerobic conditions. Growth curves for all three strains were generated and LAP expression was verified by Western blotting, and immunofluorescence staining using anti-LAP mAb[38].

**Mammalian cells.** The human colon carcinoma Caco-2 cell line (ATCC # HTB37) from 25 to 35 passages and the MDCK cell line (ATCC # CCL34) from 10 to 20 passages were cultured in Dulbecco's Modified Eagle's medium (DMEM) (Thermo Fisher Scientific) supplemented with 4 mM L-glutamine, 1 mM sodium pyruvate, and 10% fetal bovine serum (FBS; Atlanta Biologicals). The Caco-2 cells presenting stable suppression of Hsp60 mRNA (Hsp60⁻), or presenting a non-targeting control shRNA vector (Hsp60^Vec) or exhibiting a constitutive overexpression of Hsp60 were previously developed (Hsp60⁺)[32] and cultured in DMEM supplemented with 4 mM L- glutamine, 1 mM sodium pyruvate, 10% FBS, and 800 μg/ml Geneticin.

**Mice.** Female mice (A/J: 6–8 weeks of age) that are highly sensitive to oral *Lm* challenge[31,37,46] were purchased from Jackson Laboratories. Upon arrival, animals were provided ad lib feed (Rodent Diet 5001, LabDiet), sterile deionized water, and acclimatized for 5 days before the experiment. Shepherd's™ ALPHA-dri® (alpha-cellulose) was used for bedding. A cycle of 12-h artificial light and 12-h darkness was maintained. Relative humidity was 50–60% and the temperature was 20–25 °C. Mice were randomly assigned to eight different groups. Overnight cultures of each *Lactobacillus* strains (LbcWT, LbcLAP^Lm, and LbcLAP^Lin) grown in modified MRS broth were collected and centrifuged at 3500×*g* for 15 min. After three washes in phosphate-buffered saline (PBS, pH 7.4) the pellets were and resuspended in sterile deionized water at 4–8 × 10⁹ CFU/ml and replenished daily with fresh *Lactobacillus* cultures for 10 days. Naive animals received only water.

On the day of the challenge, food and water were withdrawn 5 h before oral gavage. The 12-h grown *Lm* F4244 (WT, clinical isolate) resuspended in 100 μl of PBS, (pH 7.4) containing ∼5 × 10⁸ CFU were administered orally using a stainless-steel ball-end feeding tube (Popper). The uninfected naive mice received only PBS (pH 7.4)[37]. The food was returned 1 hpi, and the mice were sacrificed 24 h (day 11) and 48 hpi (day 12, 17, and 22) using CO₂ asphyxiation. Animals were observed for clinical signs, such as ruffled hair, movement and recumbency, and their feeding and drinking habits. Body weights were also recorded during the experiments.

The animal procedure was approved by the Purdue University Animal Care and Use Committee (PACAUC approval No.1201000595) who adheres to the recommendations of the Guide for the Care and Use of Laboratory Animals published by the National Institutes of Health.

**Western blotting.** To assess the expression of LAP in the BLP strains or *Listeria* strains were grown as above. To isolate cell wall-associated proteins washed

bacterial pellets from 10 ml of overnight-grown cultures were resuspended in 0.5 ml protein extraction buffer (0.5% SDS, 10 mM Tris at pH 6.9), and incubated at 37 °C for 30 min with agitation[37]. The samples were centrifuged (14,000×$g$, 10 min, 4 °C), and the supernatants (containing cell wall-associated proteins) were retained. To isolate secreted proteins, cell-free culture supernatants were concentrated by 10% trichloroacetic acid (w/v) and centrifuged (10,000×$g$ for 20 min at 4 °C)[37]. The resulting pellet was resuspended and washed with ice-cold acetone, centrifuged, and residual acetone was evaporated. The pellet was resuspended in alkaline rehydration buffer (100 mM Tris-base, 3% SDS, 3 mM DTT, pH 11), and boiled for 10 min. To isolate total bacterial proteins (whole-cell lysates from bacterial pellets) the Bacterial Protein Extraction (B-PER) (Thermo Fisher Scientific) reagent was used.

To extract the proteins from Caco-2 cells, cells were seeded in six-well plates for 14–21 days. The cells were washed, scraped from the bottom of six-well plates, suspended in PBS, and pelleted by centrifugation. The detergent-insoluble (membrane) and the detergent-soluble (cytosolic) proteins were isolated using a Mem-Per Eukaryotic Protein Extraction Kit (Thermo Fisher Scientific). Halt proteases and phosphatase inhibitors (Thermo Fisher Scientific) were used during all protein extraction procedures. The protein concentrations were determined by BCA assay (Thermo Fisher Scientific) and separated on SDS-PAGE gels (10–12.5% polyacrylamide) and electro-transferred to the polyvinylidene difluoride (PVDF) membrane (Millipore). The membranes were then blocked in 5% nonfat dry milk (NFDM) in 0.1 M Tris-buffered saline, pH 7.5 (TBS) containing 0.1% Tween 20 (TBST) for at least 1 h. All primary antibodies were diluted in 5% NFDM in TBST and incubated overnight. Primary antibodies (Supplementary Table 1) against LAP, InlA, NamA were used at (1 µg/ml). Antibodies against Hsp60 (mAb), occludin, and MEK 1/2 were used at 1:1000 dilution, and β-actin at 1:2000 dilution. The HRP-linked secondary antibodies; anti-rabbit IgG or anti-mouse IgG (1:2000 dilution in 5% NFDM in TBST) were incubated at 37 °C for 1 h, and a chemiluminescence method was performed using LumiGLO reagent (Cell Signaling) before visualization in the Chemi-Doc XRS system (Bio-Rad). To immunoprobe, the same membrane with another antibody, the originally bound antibodies were stripped using Restore Western Blot Stripping Buffer (Thermo Fisher Scientific) according to the manufacturer's protocol. The average reaction intensities of the bands were determined using Quantity One software (Bio-Rad) and presented as the mean ± SEM after normalization to the loading control and are presented as % change relative to the control (untreated cells set at 100%). Immunoblots data are representative of three independent experiments.

**Adhesion, invasion, and translocation profiles of Listeria monocytogenes and Lactobacillus casei strains**. To analyze the adhesion or invasion profiles of $Lm$ or $L. casei$ strains (LbcWT, Lbc$^{vec}$, LbcLAP$^{Lm}$, or LbcLAP$^{Lin}$) strains, overnight bacterial cultures were washed three times with PBS and resuspended in DMEM containing 10% fetal calf serum (D10F). Bacterial cultures were then added to polarized Caco-2 monolayers (cultured for up to 14–21 days) at a multiplicity of infection (MOI) of ~10 or ~50 for $Lm$ strains or a multiplicity of exposure (MOE) of ~10 for $L. casei$ strains[32].

To measure bacterial adhesion, monolayers were rinsed in DMEM after 1 h of incubation at 37 °C in 5% CO$_2$ and were lysed with 0.1% Triton X-100 in PBS to release the adherent bacteria. The adherent bacteria (CFU) were enumerated by plating serial dilutions of the lysates on $Listeria$ selective modified oxford (MOX) agar plates for $Lm$ counts and on $Lactobacillus$ selective MRS agar plates for $L. casei$ counts.

For bacterial invasion, Caco-2 monolayers were rinsed in DMEM after 1 h of bacterial exposure as described above and incubated with D10F supplemented with gentamicin (50 µg/ml) for an additional 1 h at 37 °C in 5% CO$_2$ to kill the extracellular bacteria. Caco-2 cells were then rinsed in DMEM, lysed with 0.1% Triton X-100 in PBS and the internalized bacteria were enumerated by plating the serial dilution of the lysates on agar plates as above.

To analyze the in vitro translocation efficiencies of $Lm$ or $L. casei$ strains, Caco-2 cells were grown as monolayers on Transwell inserts with 3.0-µm pores (Corning-Costar) for up to 14–21 days[32]. The integrity of monolayers was monitored by measuring the TEER (Millicells Voltmeter, Millipore) and at least 200 Ω/cm$^2$ (±10) was used as the standard. Overnight bacterial cultures were washed with PBS, resuspended in D10F, and were added to the apical side of the Transwell at an MOI of ~50 for $Lm$ and MOE of ~10 for $L. casei$ strains. After 2 h incubation at 37 °C in 5% CO$_2$, the liquid from the basal well was serially diluted and the translocated bacteria were enumerated by plating. The percent of bacteria adhered, invaded to or translocated across the Caco-2 monolayers was calculated: (the number of viable bacteria recovered at the basal side/The number of viable bacteria added to the apical side) × 100.

**Immunofluorescence staining of live bacterial cells**. The expression of LAP in the BLP and $Listeria$ strains was assessed by immunofluorescence staining. Briefly, the stationary phase-grown (at 37 °C) cultures of $L. casei$ in mMRS or $Listeria$ in TSBYE (0.6%), were harvested by centrifugation, washed twice with cold PBS, and incubated with the mouse anti-LAP mAb-H7 (1:50 in 5% BSA in PBS) at 37 °C for 1 h and washed in cold PBS[70]. Bacterial cells were then incubated with Alexa-488-conjugated goat anti-mouse secondary antibody (Cell Signaling, 1:50 in 5% BSA in PBS) at 37 °C for 1 h, washed at least four times with 0.5% Tween 20 in PBS and

viewed in a Nikon A1R confocal microscope (equipped with 405-nm/Argon/561-nm lasers) using a Plan APO VC 60X/1.40 NA oil immersion objective with the Nikon Elements software (Nikon Instruments Inc.).

**Inhibition of Listeria monocytogenes adhesion, invasion, and translocation by L. casei strains**. To investigate the ability of $L. casei$ strains to inhibit the adhesion, invasion, and translocation, Caco-2 (Caco-2, Hsp60$^{Vec}$, Hsp60$^-$ or Hsp60$^+$) were used[38]. MDCK cells were cultured for 4–5 days. Briefly, overnight cultures of $L. casei$ strains (LbcWT, Lbc$^{vec}$, LbcLAP$^{Lm}$, or LbcLAP$^{Lin}$) were washed three times with PBS, resuspended in D10F and were added to each well of cultured Caco-2 or MDCK cells (MOE ~10) and incubated for 24 h at 37 °C in 5% CO$_2$. Unbound bacteria were removed by rinsing monolayers with DMEM. Overnight-grown $Lm$-WT (F4244) cultures were washed with PBS, resuspended in D10F was added to Caco-2 or MDCK cells (MOI ~50) and incubated for 1 h at 37 °C with 5% CO$_2$. The cell monolayers were then rinsed three times with DMEM and lysed with 0.1% Triton X-100 in PBS. The adhered bacterial counts were determined as above.

To investigate the ability of $L. casei$ strains to inhibit the invasion of $Lm$, $L. casei$ strains were added to Caco-2 or MDCK cells (MOE~10, Caco-2, Hsp60$^{Vec}$, Hsp60$^-$, or Hsp60$^+$) for 24 h and then infected with $Lm$ (MOI ~50) as above. To determine the intracellular $Lm$ counts, the cell monolayers were incubated in D10F supplemented with gentamycin (50 µg/mL) to kill the extracellular bacteria for an additional 1 h at 37 °C with 5% CO$_2$. Epithelial cells were lysed, and the invaded bacteria were enumerated.

To investigate the ability of $L. casei$ strains to inhibit the translocation of $Lm$, the Caco-2 (Caco-2, Hsp60$^{Vec}$, Hsp60$^-$, or Hsp60$^+$) or MDCK monolayers grown on Transwell filter inserts as above. $L. casei$ cultures were first added to the apical wells (MOE, 10) and incubated at 37 °C with 5% CO$_2$ for 24 h. Unbound bacteria were removed by rinsing cells with DMEM. Overnight-grown $Lm$ culture resuspended in D10F were added to epithelial monolayers (MOI ~50) and incubated at 37 °C for 2 h with 5% CO$_2$. The liquid from the basal well was removed, serially diluted, and plated on MOX agar plates to enumerate $Lm$. The percent of $Lm$ adhered, invaded to or translocated across cell monolayers was calculated as above.

**Antimicrobial activity**. To assess the antilisterial activity of $L. casei$ cultures, Petri plates were poured with sterile brain heart infusion (BHI; Neogen) agar (1.5% agarose) and overlaid with sterile BHI soft agar (0.8% agarose) seeded with 20 µl of freshly (12 h) grown $Lm$ F4244. Wells of 7.0-mm diameter was dug aseptically with a cork borer and 100 µl of $L. casei$ cultures (LbcWT, Lbc$^{vec}$, LbcLAP$^{Lm}$, or LbcLAP$^{Lin}$) grown to a stationary phase in mMRS broth at 37 °C were loaded per well. As a positive control, 100 µl of $Pediococcus acidilactici$ strain H (pediocin producer) grown to a stationary phase in MRS broth at 37 °C and 100 µl of van-comycin (100 mg/ml) were loaded per well. Plates were kept at 4 °C for 15 min for absorption and incubated at 37 °C for 24 h to observe the zones of inhibition.

**Growth characteristics of BLP strains in growth media and artificial gastro-intestinal fluids**. For growth curve experiments, the $L. casei$ strains (LbcWT, Lbc$^{vec}$, LbcLAP$^{Lm}$, and LbcLAP$^{Lin}$) were inoculated (1%) in MRS or mMRS containing the appropriate antibiotics and incubated at 37 °C and absorbance at 600 nm was determined at 2-h intervals for a period of 12 h and at 24 h by using a microplate reader (Bio-Rad). The assay was repeated three times with duplicate samples.

The survival of $L. casei$ strains exposed sequentially to the simulated gastrointestinal fluid (SGF) and simulated intestinal fluid (SIF-I and SIF-II, to simulate gastric phase, enteric phase 1 and enteric phase 2, respectively), over 6 h (2 h for each step) period was monitored by plating. SGF contained pepsin (3 g/L) and lipase (0.9 mg/L) (Sigma-Aldrich), pH 1.2–1.5 (adjusted using 1 N HCl). Both SIF-I and SIF-II contained bile (bovine bile; 10 g/L, Sigma-Aldrich) and porcine pancreatin (1 g/L; Sigma-Aldrich), but SIF-I pH was 4.3–5.2 and SIF-II pH was 6.7–7.5 (adjusted using alkaline solution; 150 ml of 1 N NaOH, 14 g of PO$_4$H$_2$Na.2H$_2$0 and deionized water up to 1 L).

Overnight cultures of $L. casei$ strains were washed and resuspended in SGF (100 ml) and incubated at 37 °C, with agitation (150 rpm for 2 h) (gastric phase), and bacterial counts were monitored every 30 min for 2 h. The cells from SGF were pelleted and transferred sequentially into SIF-I, and SIF-II, incubated each at 37 °C for 2 h to simulate the initial and final phases of intestinal digestion. $L. casei$ counts were enumerated on MRS agar plates and the assay was repeated three times with duplicate samples. The level of LAP expression in BLP cultures following exposure to SGF and SIF (I and II) was also monitored by western blotting using anti-LAP mAb. The survival of BLP strains in water is also ensured during animal treatment in a 24-h cycle.

**Enumeration of L. monocytogenes and L. casei in mouse organs and samples**. Mice were euthanized by CO$_2$ asphyxiation at 10 days following probiotic treatment or 24 (day 11) and 48 (day 12, 17, 22) hpi, and the intestine (duodenum, jejunum, ileum, cecum, and colon), blood, MLN, spleen, liver, kidney were aseptically collected.

To assess the $L. casei$ loads in the lumen, the entire intestinal contents were removed and homogenized using a tissue homogenizer (Bio Spec) in 5 ml of PBS

containing 0.1% Tween 20 (PBS-T). To assess the mucosal-associated bacteria (*Lm* or *L. casei*), the entire length of the intestine (duodenum–colon, for *Lm*) or the segments of the intestine (ileum, cecum, and colon, for *L. casei*) was flushed with ice-cold sterile PBS that removed the luminal contents and loosely adherent bacteria. The whole intestine was homogenized in 9 ml of buffered *Listeria* enrichment broth (BLEB) containing 0.1% Tween 20 and selective antimicrobial agents (Neogen) to enumerate *Lm*. The intestinal segments were homogenized in 4.5 ml PBS-T to enumerate *L. casei*. To assess invaded bacterial counts (*Lm* or *L. casei*), intestinal segments (ileum, cecum, and colon) were treated with gentamicin (100 μg/ml) in DMEM for 2 h at room temperature to kill extracellular bacteria. After five washes in DMEM, the intestinal segments were homogenized in 1 ml of PBS-T. Fecal pellets were weighed, and then suspended in PBS (150 mg/ml) and homogenized using sterile toothpicks.

Aseptically harvested extraintestinal organs/tissues were homogenized using a tissue homogenizer in 4.5 ml (spleen, kidney, and MLN) or 9 ml (liver) of BLEB containing 0.1% Tween 20 and selective antimicrobial agents (Neogen) for *Lm* counts and in PBS-T for *L. casei* counts. To enumerate *Listeria*, LbcWT and BLP strains (LbcLAP$^{Lm}$ and LbcLAP$^{Lin}$), the tissue and fecal homogenates were serially diluted in PBS and plated on MOX agar plates containing selective antibiotics (Neogen), MRS agar containing vancomycin (300 μg/ml), and MRS agar containing vancomycin (300 μg/ml) and erythromycin (2 μg/ml), respectively. No colonies were detected on MRS + vancomycin plates from mock-treated (naive) animals or animals that received *Lm* only.

Blood was collected using a 1 ml syringe with a 21-G needle by cardiac puncture. To enumerate the burdens of *Listeria* in the blood, 50 μl of blood was diluted with 450 μl of BLEB immediately following collection and samples were serially diluted and plated as above. In specific experiments, a section of the ileum (~2 cm) or colon (~2 cm) was fixed overnight in 10% formalin for histopathology or immunostaining.

**Clinical sign score**. After infection, mice were monitored and scored[3] blindly by a veterinarian for disease severity by two parameters: weight loss (>95% of initial weight = 0, 95–90% initial weight = 1, 90–80% initial weight = 2, and <80% initial weight = 3) and morbidity (score of 0–5 for each symptom; labored breathing, response to external stimuli, movement, recumbency, and ruffled fur).

**Immunohistochemistry, Alcian blue staining, and histopathology**. Mouse tissues were fixed in 10% neutral buffered formalin for 24–48 h, placed in a Sakura Tissue-Tek VIP6 tissue processor for dehydration through graded ethanol (70%, 80%, 95%, and 100%), cleaned in xylene, and embedded in Leica Paraplast Plus paraffin. Tissue sections (4 μm) were made using a Thermo HM355S microtome. Sections were mounted on charged slides and dried for 30–60 min in a 60 °C oven. All slides were deparaffinized through three changes of xylene (5 min each) and rehydrated through graded ethanol as above in a Leica Autostainer XL. Slides are stained in Gill's II hematoxylin blue and counterstained in an eosin/phloxine B mixture using the Leica Autostainer XL. Finally, slides were dehydrated, cleared in xylene, and mounted with coverslips in a toluene-based mounting media (Leica MM24).

For immunohistochemistry, after deparaffinization, antigen retrieval was done in the appropriate buffer using a BioCare decloaking chamber at 95 °C for 20 min. Slides were cooled at room temperature for 20 min dipped into TBST. The rest of the staining was carried out at room temperature using a BioCare Intellipath stainer. Slides were incubated with 3% hydrogen peroxide in water for 5 min, or Bloxall block for 10 min for antibody labeling. Slides were rinsed with TBST and incubated in 2.5% normal goat or horse serum for 20 min. Excess reagents were removed, and a primary antibody or antibody cocktail was applied at the appropriate dilution for 30 min. Primary antibodies (Supplementary Table 1) include antibodies to *Listeria* (1:100 dilution), ZO-1 (1:100 dilution), LAP (1:1000 dilution), Muc2 (1:500 dilution), Ki67 (1:100 dilution), cleaved caspase-3 (1:200 dilution), IL-10 (1:100 dilution), TGFβ (1:250 dilution), CD3 (1:200 dilution), CD4 (1:100 dilution), CD8α (1:1000 dilution), F4/80 (1:200 dilution), FoxP3 (1:200 dilution), CD11c (1:500 dilution), and NKp46 (1:100 dilution). Negative control slides were stained with their respective isotype controls (Supplementary Table 1) at 1–2 μg/mL for 30 min. After TBST rinse (twice) the secondary antibody was applied for 30 min, rinsed (twice) in TBST before reaction with Vector ImmPACT DAB (Vector Labs) for 5 min. Slides that were probed with two antibodies were counterstained with ImmPACT Vector Red (Vector Labs). Slides were rinsed in water and counterstained with hematoxylin. Tissue sections were also stained with Alcian blue for goblet cell counts.

For histopathology, a microscopic examination was performed by a board-certified veterinary pathologist who was blinded to the treatment groups and the interpretation was based on standard histopathological morphology. The extent of mouse ileal lesions was determined by using a semi-quantitative method that included the amount of inflammatory infiltrate and the percentage of goblet cells comprising the villous epithelium. The mouse ileal tissues were scored on a scale of 0–3 for the aforementioned two parameters yielding a maximum score of 6. A histomorphological scale for assessing inflammation in the lamina propria of the mucosa is provided as follows: 3, marked amounts (sheets of granulocytes expanding the width of the villous tip); 2, moderate amounts (sheets of granulocytes at the base of the villous); 1, mild amounts (multifocal scattering); and

0, none observed. To estimate percentage of goblet cells loss, following scale was used: 3, 50% or greater; 2, 25–50%; 1, 11–25%; and 0, <10%.

For H&E-stained and immunoperoxidase-stained tissues were imaged using a DMLB microscope (Leica) with ×40/0.25 NA HC FL PLAN or a ×100/1.40 NA HC FL PLAN oil immersion objective and a DFC310 FX (Leica) camera controlled by Leica Application Suite. Post-acquisition processing, including the stitching of tiled images, was performed using Leica Application Suite (Leica). Immunoperoxidase-stained positive cells such as immune cell infiltrate were counted manually on tiled images in a blinded manner. For each experiment, immunoperoxidase-stained positive cells from 25 villi in the tissue sections of three to four individual animals per treatment were recorded. Each point represents an average of 25 villi from a single mouse.

**Survival study**. For the survival study, a single LD$_{50}$ dose (estimated by preliminary experiments) of ~2.5 × 10$^9$ CFU/mouse (100 μl volume) was administered per oral route using a stainless-steel ball-end feeding tube (Popper) and observed for 10 days and mortality was recorded. Animal body weight was also recorded, and mice were sacrificed and considered deceased, if weight loss was greater than 25% for two consecutive measurements, as mandated by PACUC regulation.

**Competitive exclusion of *L. monocytogenes* by *L. casei* strains**. Bacterial cultures were prepared as above and were suspended in D10F to a final concentration of 1 × 10$^7$ CFU/ml. For competitive adhesion, *Lm* was co-inoculated with each of the *L. casei* strains (LbcWT, Lbc$^{vec}$, LbcLAP$^{Lm}$, or LbcLAP$^{Lin}$, 1:1 ratio) to Caco-2 cell monolayer to achieve an MOI/E of 50 and incubated at 37 °C for 1 h with 5% CO$_2$. The cell monolayers were rinsed three times with D10F and lysed with 0.1% Triton X-100 in PBS. Adherent bacteria were enumerated by plating the serial dilution of resulting lysates on *Lactobacillus* selective MRS agar and *Listeria* selective MOX agar plates. The percent of bacteria adhered to cells was calculated as described above.

**Analysis of *L. casei* and *L. monocytogenes* co-aggregation in vitro**. *Lm* F4244, *Lin* F4248, LbcWT, LbcLAP$^{Lm}$, and LbcLAP$^{Lin}$ were cultured at 37 °C for 16–18 h in TSBYE (*Listeria* strains) or mMRS (*L. casei* strains). All cultures were pelleted by centrifugation at 8000 × *g* for 3 min and washed with sterile PBS. All cultures were serially diluted to obtain a cell concentration of 10$^6$ CFU/ml. *Lm* or *Lin* cultures were allowed to interact with the individual probiotic strains (LbcWT, LbcLAP$^{Lm}$, or LbcLAP$^{Lin}$) at a 1:1 ratio in PBS for 1 h at room temperature with constant agitation on Lab Doctor Revolver (Mid Scientific). Anti-Listeria magnetic Dynabeads (Thermo Fischer Scientific) were used to capture and separate *Lm* from unbound probiotics. Briefly, 20 μl/ml of bead slurry was added to the bacterial mixtures and allowed to interact for 10 min at room temperature with constant agitation. Beads were magnetically separated and washed with sterile PBS-T three times (10 min each wash) with constant agitation. Beads were serially diluted and plated on MOX and MRS agar for enumeration of *Listeria* and *Lactobacillus*, respectively. For blocking the surface-expressed LAP on the BLP strains, all *L. casei* strains were harvested from 1 ml of overnight-grown culture, and the pellets were washed three times in PBS before the addition of 5 μg/ml of mouse-monoclonal anti-LAP mAb or the isotype control mouse IgG (Santa Cruz Biotechnology). The mixture was incubated at 37 °C for 1 h with gentle shaking and then pelleted, washed five times in the PBS, and used in the aforementioned capture assay. Bead-captured bacteria were also examined under confocal fluorescence microscopy. Images were acquired using a Nikon A1R confocal microscope (equipped with 405-nm/Argon/561-nm lasers) using a Plan APO VC ×60/1.40 NA oil immersion objective with the Nikon NIS Elements software (Nikon Instruments Inc.).

**Immunofluorescence staining and confocal microscopy**. The mouse ileal or colonic tissue sections were fixed with 10% formalin and embedded in paraffin. The tissues were sectioned (5-μm thick), deparaffinized, and rehydrated for antigen retrieval by immersing the slides in boiling sodium citrate buffer (10 mM, pH 6.0) or 0.01 M Tris/EDTA (pH 9.0), for 10 min. The tissue sections were permeabilized and blocked with PBS containing 0.3% Triton X-100 (Sigma-Aldrich) and 3% normal goat serum (Cell signaling) and immunostained with specific primary antibodies or a cocktail of primary antibodies (Supplementary Table 1) by incubating overnight at 4 °C. Primary antibodies included antibodies to MLCK (1:100 dilution), P-MLC (1:200 dilution), claudin-1 (1:200 dilution), occludin (1:150 dilution), E-cadherin (1:200 dilution), p65 (1:800 dilution), and P-p65 (1:100 dilution). Slides were then rinsed with PBS (three cycles, 5 min), and were incubated with respective Alexa Fluor 488/555/647-conjugated secondary antibody (1:500 dilution) for 2 h at room temperature followed by 3× washing with PBS (5 min each). The nuclei were stained with DAPI (500 ng/ml; Cell signaling), and slides were mounted in ProLong antifade reagent (Invitrogen).

For antibody labeling in cells, Caco-2 cells were grown to 40–50% confluence in eight-chambered slides (Millipore). At the end of the treatment, the cells were fixed with 3.7% formaldehyde in PBS for 20 min and permeabilized and blocked with PBS containing 0.3% Triton X-100 and 3% BSA (Sigma-Aldrich) for 1 h at room temperature and then incubated with primary antibodies (Supplementary Table 1) to LAP (1:50 dilution), ZO-1 (1:100 dilution), (Hsp60 pAb, 1:100 dilution) or similar primary antibodies or a cocktail of primary antibodies at dilutions

described above overnight at 4 °C. The cells were then washed with PBS (three cycles, 5 min) and incubated with respective Alexa Fluor 488/555/647-conjugated secondary antibodies (1:500 dilution) for 2 h at room temperature. The nuclei were stained with DAPI (500 ng/ml; Cell Signaling) and slides were mounted in ProLong antifade reagent (Invitrogen).

All images were acquired using a Nikon A1R confocal microscope as above. The X-Z and Y-Z cross-sections were produced by orthogonal reconstructions from z-stack scanning at 0.15-μm intervals taken with ×60 objective in a 5-μm-thick paraffin-embedded tissue section or Caco-2 monolayers. Three-dimensional reconstructions were performed using Nikon NIS Elements software (Nikon Instruments Inc.). Post-acquisition processing was done in Adobe Photoshop.

The p65 and P-p65 nuclear-positive cells were counted and expressed as average nuclear-positive cells per 15 villi from four individual mice per treatment. For quantification of MLCK and P-MLC expression or analysis of redistribution of cell–cell junctional proteins, images of Caco-2 cells or mouse ileum from five different fields from three independent experiments (for Caco-2 cells, representing 90–100 cells) or three individual mice (representing 100–150 epithelial cells) per treatment were acquired. The relative expression levels of MLCK and P-MLC were analyzed by using the NIH ImageJ software. For the analysis of redistribution of cell–cell junctional protein or *Lm* cells adhered, % of the total number of cells containing intracellular cell–cell junctional protein puncta and the number of *Lm* cells adhered were manually counted in acquired images and calculated, respectively.

**Giemsa staining**. To assess bacterial adhesion, Caco-2 cells were grown to 40–50% confluence in eight-chambered slides (Millipore). At the end of the treatment, cell monolayers were stained with Giemsa stain followed by microscopic examination to visualize bacterial attachment qualitatively. Images were acquired using a DMLB microscope (Leica) with a ×100/1.40 NA HC FL PLAN oil immersion objective.

**Fluorescent in-situ hybridization (FISH)**. After deparaffinization, the enzyme digestion and hybridization were carried out on a BioCare Medical IQ Kinetic slide stainer. Slides were incubated with proteinase K (Dako) at 20 μl/ml in TBST at 37 °C for 10 min, washed with 2× saline sodium citrate (SSC) buffer containing 0.1% Tween 20, dehydrated in graded ethanol (70%, 95%, and 100%) and air-dried. Slides were then incubated with Alexa-488-conjugated-Lcas467 *L. casei* DNA probe (5′/5Alex488N/CCGTCACGCCGACAACAG-3′) at a 1:60 dilution (0.11 ng/μL) (Supplementary Table 1). The oligomer was denatured at 78 °C for 5 min and then hybridized at 45 °C for 16 h. Slides were washed with 2× SSC, stringency wash with 0.1× SSC carried out at 60 °C, and washed again with 2× SSC. Finally, slides were counterstained with DAPI for 5 min at 1 μg/mL and mounted with coverslips with Prolong Gold. Images were acquired using a Nikon A1R confocal microscope as above.

**Biofilm assay**. The microtiter plate biofilm assay was used to quantify biofilm formation with slight modifications[71]. Briefly, the optical density ($OD_{595}$ nm) of overnight cultures of *Lm*, *S. typhimurium*, and *L. casei* (LbcWT, Lbc[vec], LbcLAP[Lm], or LbcLAP[Lin]) was adjusted to 1.2. The cultures were diluted 40-fold in a mixture of (1:1 ratio) brain heart infusion and mMRS broths. One hundred and fifty microliters of diluted monocultures or co-cultures (mixed prior in 1:1 ratio) were aliquoted into wells of a 96-well tissue culture-treated microtiter plate (Corning) and incubated at 32 °C for 48 h. Following incubation, the supernatant from each well was aspirated to remove loosely attached cells, and the wells were washed three times with 10 mM PBS. The microtiter plate was air-dried for 15 min and 150 μl of 0.1% crystal violet (CV) solution was added to each well and incubated for 45 min at room temperature to stain the biofilm cells. Each well was washed four times with sterile water to remove residual CV stain and air-dried for 15 min. Two hundred microliter of 95% ethanol was added to each well and incubated for 15 min at room temperature to destain the biofilm. Finally, the ethanol solution from each well was transferred to a fresh flat-bottom microtiter plate and absorbance at 595 nm was measured. Wells were also imaged before the addition of ethanol.

**Analysis of epithelial cell–cell junctional integrity**. To test the effect of *L. casei* strains on cell–cell junctional integrity, the TEER of Caco-2 cells before and after treatment was measured using a Millicell ERS system (Millipore). For analysis of epithelial permeability, 5 mg/ml of 4 kDa FITC-Dextran (FD4; Sigma) in D10F was added to the well (apical side) and translocation of FD4 to the basal side was monitored by a spectrophotometer (Spectramax, Molecular Devices).

**Intestinal permeability assay**. Four to five hours before sacrifice, mice were orally gavaged with 100 μl of FD4 (15 mg/100 μl, Sigma)[31]. Urine voluntarily excreted during euthanasia, was collected from the tray/bag, and blood was collected by cardiac puncture. Sera and urine were appropriately diluted in PBS and assayed for FD4 by measuring in a spectrophotometer (excitation: 485 nm; emission 520 nm; Molecular devices) The FD4 concentration was calculated using a standard curve generated by serially diluting FD4 in PBS.

**Cytokine ELISA**. The quantification of TNFα, IL-6, and IFNγ protein levels was performed in the ileal tissue lysates homogenized in cell lysis buffer. The protein concentrations in the ileal tissue lysates were determined by BCA assay and equal amounts of protein were assayed using mouse-specific ELISA kits (Ray Biotech) as per the manufacturer's instruction.

**Statistical analysis**. Experimental data were analyzed using Microsoft Excel and GraphPad Prism (La Jolla, CA) software. *P* values and the type of statistical analysis performed are described in the figure legends. For mouse microbial counts, statistical significance was assessed by the Mann–Whitney test. For the mice survival experiment, the Kaplan–Meyer plot was generated, and a log-rank test was performed. In other experiments, comparisons between treatment and control were performed using the one-way or two-way analysis of variance with Tukey's multiple-comparison test. Unless otherwise indicated, data for all experiments are presented as the mean ± standard error of the mean (SEM).

**Reporting summary**. Further information on research design is available in the Nature Research Reporting Summary linked to this article.

## Data availability

All relevant data are available from the corresponding author upon request. Source data are provided with this paper.

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

## Acknowledgements

A part of the research was supported by the funds from the Purdue University AgSEED program, BioMatrix, Inc., the Agricultural Research Service (ARS) of USDA (1935-42000-072-02G), the USDA National Institute of Food and Agriculture (Hatch), and Purdue Research Foundation the Purdue Research Foundation, the Purdue University Office of Executive Vice President for Research and Partnerships (EVPRP), and the Postdoctoral challenge grant from the Indiana Clinical and Translation Sciences Institute (NIH- sponsored; awarded to RD). The authors acknowledge N. Gallina, E. Eser,

Z. Chen, T. Qi, M. Samaddar, M. Mathipa, B. Kinnamon, C. Coakley, Z. Gao, V. Nathan, T. Bailey, A. Pires dos Santos, D. Pons, and R. Mino for assistance with the animal study. We also thank Dr. Nathan Horn for helpful discussions, Dr. Suresh Mittal for sharing the MDCK cell line and Dr. Mike Miller, University of Illinois (Urbana) for providing *L. casei* ATCC334 strain.

## Author contributions

A.K.B. conceived the idea. R.D. and A.K.B. conceptualized the study. R.D., M.A.R.A., V.R., R.V., and A.K.B. designed the experiments. R.D., M.A.R.A., V.R., S.T., A.K.S, D.L., X.B., L.X., A.P.D., V.P.C., J.A.S., B.M.A., and A.K.B. performed the experiments. R.D. and A.K.B. interpreted the data. R.D. and A.K.B. wrote the paper. R.D. and A.K.B. prepared the figures. A.D.C. assisted with histopathology scoring.

## Competing interests

A.K.B., R.D., and M.A.R.A. have filed a patent covering the potential application of BLP. The remaining authors declare no competing interests.
