## [Peer Review File · Nature Communications]

REVIEWER COMMENTS

Reviewer #1 (Remarks to the Author):

The manuscript by Drolia et al is a comprehensive study of the effects of engineered probiotics on *Listeria monocytogenes* pathogenesis. The work provides a number of new insights and advancements for the field. These include (1) the description of a novel approach for engineered probiotics; (2) insights into the likely mode(s)-of-action of such probiotics and (3) insights into *Listeria* pathogenesis. (4) As the engineered (BLP) strains also show enhanced colonization of the mouse gut the work also suggests an approach to enhance engraftment of probiotics in the gut through rational engineering.

Main comments

In many cases the authors use both in vivo analyses supported by in vitro (CACO-2) studies to investigate mechanisms and the data are convincing. However in other cases (e.g. the goblet cell work - Fig4) this is not possible & the authors rely on in vivo analysis alone. This results in a difficulty in explaining the phenomenon examined whereby BLP reduces both Lm numbers and loss of goblet cells. In this scenario it is not clear whether the preservation of goblet cells/barrier function during infection is a function of the activity of BLP on the host or a phenomenon related to lower numbers of Lm in the gut. I'm not sure it is possible to state (l309-310) that BLP 'limits Lm-induced loss of mucus-producing GCs' as BLP may actually be reducing numbers of Lm in the gut by other mechanisms and it is the resulting lower numbers of Lm that reduce the phenomenon of GC loss. I am suggesting that the authors address this in the language used, to introduce this caveat into a revised text. This is also true for the data in Fig 5 - though the authors conclusions are better supported here by in vitro studies. Still I feel it is an issue that should be addressed by a statement in the revised text.

Much of the histology work is supported by quantitative data/scores (e.g. in Fig 4). However where there is no quantitation it would be useful to see some indication of number of fields that were examined per sample in order for the reader to appreciate how robust is the evidence for particular phenomena. This applies in particular to the studies of co-localisation/co-aggregation. apologies if I have missed it in the text, but I feel that it should be made more obvious to the reader and quantified as much as possible.

A strength of the work is that the BLP strains demonstrate better colonisation of the gut than the wild-type strain. Whilst I appreciate that this is a mouse study, the data may be useful for enhancing engraftment of probiotics in humans - as this is a significant issue in the field of probiotic research (where many strains are seen to be transient in the gut & therefore ineffectual). On the other hand

the genetic manipulation of gut commensals (as opposed to *L. lactis*, which doesn't colonise the gut) is controversial as such strains may colonise for the life of the host leading to issues of GMM release etc. Whilst the Introduction to the paper mentions one of these issues I feel they could be expanded in the Discussion slightly to contextualise the work. Whilst the paper is very well written I feel that the Discussion reads slightly as a re-iteration of the Results section (which is already very comprehensive). I feel that the authors should better emphasize some of the biotech implications of the work in the Discussion.

L45-46: related to the above point the authors also suggest in the Introduction that it is proposed that probiotics compete with pathogens for adhesion sites - and a citation is given to a review article. However I am not sure that good evidence exists for this. So a strength of the current work is support for this concept & greater general discussion of this point would add to the quality of the Discussion section.

Minor comments:

The following sentences do not scan particularly well and could be revised for clarity:

L59-63

L126-129

L421-424

L449-452

L501-503

Did the authors measure plasmid retention in the engineered strains?

Ref 26. To my knowledge there is an updated review from this group that would be more appropriate here: *Front Cell Infect Microbiol.* 2014 Feb 5;4:9.

Reviewer #2 (Remarks to the Author):

Drolia et al have followed up on their previous work which identified LAP as an additional receptor for *Listeria monocytogenes* (Lm) entry into the intestine, and applied this finding to engineering a probiotic strain of *Lactobacillus* that expresses the LAP receptors from either Lm or *Listeria innocua*. The paper is well done and impactful, in particular for the probiotic field in which it has been a challenge to establish colonization that would be lasting and effective. I have a couple of suggestions that I think would broaden the manuscript's impact and some areas that would require additional experiments to substantiate the authors' conclusions.

Major experimental suggestions:

The probiotic and anti-Listerial effects have been extensively characterized in two experimental models that are particularly permissive to *Listeria* infection (Caco-2 cells and A/J mice). The authors performed a very comprehensive set of experiments to characterize that BLP strains prevent Lm infection, downstream signaling, loss of barrier integrity, increased goblet cell proliferation and immune responses. It would extend and generalize the findings of the manuscript to show the same effect on Lm CFUs in C57B6 mice and/or in a less permissive cell line for infection that can form better tight junctions (i.e. the cell line that has been used historically by many groups to assess entry phenotypes MDCK cells which are known to create extremely tight epithelial barriers that replicate the situation *in vivo*). It would be excessive to ask for all of the experiments to be replicated but assessing bacterial adhesion and invasion in less permissive cells lines or mouse models would be sufficient to increase the potential scope of the findings in a a more stringent biological system.

The authors hypothesize that the BLP strains occupy HSP60 and thus outcompete Lm binding as one potential mechanism of entry however in principle you would still get binding to E-cadherin and C-met that would allow entry to the tips of villi and near goblet cells under these conditions. If HSP60 binding is occluded the authors could demonstrate that in Caco2 cells using an HSP60 antibody prior to Lm infection as a second experimental demonstration of their hypothesis. Based on their data, I was more interested in the chains of BLPs that form and seem to trap or bind to Lm in the mucus and the intestinal lumen. Does this binding occur using a biofilm assay in *in vitro* planktonic co-culture between BLP and *Listeria monocytogenes* with and without LAP (e.g. crystal violet binding to test tubes as is the case for ActA)? Are the chains formed in the intestine or normally during growth by virtue of LAP overexpression? The microscopy seems to indicate longer chains for BLPs with LAP than LbcWT but that could be based on the one representative image that is shown. Would other gastroenteric pathogens like *Salmonella* be trapped as well physically and restricted from entry or would it be unique to Lm cell-wall binding of LAP?

The timeline of colonization in the mouse model means that there would have still been BLPs in the system when Lm is administered. Would there still be BLP colonization following a longer gap prior

to infection say one or two weeks? More specifically how lasting is the BLP colonization and the protective effect it confers or would it require continuous BLP administration? This gets at the question of colonization of the mucosa versus trapping and clearing Lm through the chains and could also be assessed by Lm shedding in the feces over time (beyond the 24 and 48 hours shown) and how that is altered by BLPs. Also why would there be less Lm shed if you inoculate the same amount and get less infection? Wouldn't it follow that the bacteria would be more shed into the feces originally? Do you need to add a Day 1 time point to see the majority of the Lm that is shed into the feces? Where else would it go?

Minor suggestions:

Fractionation controls are missing— for both the HSP60 localization and the Listeria/BLP cell wall fractionation experiments the fractions should be run on the same blot with controls that indicate that the fractionation worked (i.e. for Listeria InIA, not sure what would be appropriate for Lbc) or a membrane/TJ marker for Caco2. Since the localization of LAP on L.inocua is discussed it should be included in the fractionation experiments to show that it does not attach to the cell wall as it does for Lm. This was shown with fluorescent imaging but should be demonstrated with WB like it is for Lm. In addition, if it is secreted and released this could be shown through a TCA precipitated secreted fraction a control for this could be p60.

The histology used throughout is very nice but it would be good to have zoomed out images that show the gross distribution of Lm and BLP to demonstrate distribution throughout the intestine and show that infected vili were not specifically sought out.

Additionally, the authors undertook a heroic effort in quantifying immune cells by histology but wouldn't FACS be more quantitative and representative of immune infiltrate into the total tissue?

Reviewer #3 (Remarks to the Author):

The manuscript by Drolia et describes that the pre-colonization of Listeria monocytogenes (LM)-susceptible A/J mice with Lactobacillus casei expressing the LM surface protein LAP (LcbLAP) probiotics decreases the severity of LM infections. The authors report lower numbers of invasive bacteria in the guy and distal organs, as well as improved clinical scores and survival. The effects of LcbLAP are attributed to three main mechanisms: (1) competitive exclusion of LM adhesion to

epithelial cells; (2) improved intestinal barrier function; and (3) contact-dependent immune-modulation.

The use of probiotics to prevent or mitigate LM infection has been of recent interest, however most of these studies do not utilize LM protein-producing engineered probiotic strains. Additionally, these studies tend to focus on the production of bacteriocins and other competitive exclusion mechanisms, whereas the present study outlines multiple functions of the engineered LcbLAP strain: (1) competition, (2) barrier integrity, and (3) immuno-modulation. A previously published paper regarding the use of LcpLAP in preventing LM infection looked at adhesion, invasion, and translocation, as well as probiotic induced anti-LM cytotoxicity. The earlier strain did not show as promising results for prevention of dissemination of LM infection. This paper looked at potential mechanisms of LcbLAP for mitigating lethal LM infection.

Overall, the data presented appears to convincingly support the premise that this engineered probiotic strain can help reduce severe LM infection. What is unclear is the actual utility of this approach – it would appear that the probiotic may need to be continually consumed to offer protection, and it seems unlikely that probiotics would have much effect after LM has successfully crossed the intestinal barrier, and therefore are unlikely to represent a treatment (vs a preventative). This caveat might be mitigated if it was demonstrated that the probiotic provided some protection when not consumed daily – if it was able to stably colonize the GI tract.

Specific comments:

1. Figure 1: Data from Figure 1 was obtained using standard techniques to monitor adhesion, invasion, and translocation of bacteria. Data are presented as mean +/- SEM; statistical analysis (one or two way ANOVA with Tukey's multiple comparison) is appropriate.

Adhesion/invasion/translocation of Lap- LM after LcbLAP treatment would be a nice control to compare how important Lap interaction is in preventing adhesion/invasion/translocation.

2. Line 136: a 100% increase in adhesion is essentially a two-fold increase.

3. Figure 2: CFU data shown as mean +/- SEM. The median is less affected by outliers vs SEM and is preferable. How stable is LcbLATPLm colonization? Does protection require continuous feeding of the probiotic?

4. It is unclear if statistical significance is measured between each group individually, or clumped together (ie naïve vs LcbWt vs LcbLAPLM vs LcbLAPlim or naïve + LcbWT vs LcbLAPLM + LcbLAPlim). Statistical tests used are appropriate, but data being compared could be clarified.

5. Supp Figure 2: Is part H necessary? It is hard to distinguish major differences between mice (some ruffling is evident, but weight loss, breathing changes, and movement can't be discerned from photographs).

6. Figure 3: Anti-LM microbeads were used to capture LM and co-capture Lcb. Is it possible that the anti-LM immunobeads could recognize/capture LM-LAP protein expressed on Lcb? This experiment demonstrated relatively equal numbers of LM and LcbLAP captures. Immunofluorescence and IHC show colocalization of Lcb and LM and Lcb and Lin in vivo.

7. Supp Figure 3: Colocalization of Lcb with Hsp60. Anti-Hsp60 antibody inhibition of binding by LcbLAP is a nice touch. Immunoprecipitation of LcbLAP with Hsp60 might also be valuable to show?

8. Supp Figure 4: Alcian blue staining shows change in number of goblet cells. This is enumerated by graph. It would be nice to see if there is statistical significance between LcbLAP with and without infection.

9. Fig. 5E: Cell membrane expression of junction proteins in LcbWT + LM looks very different from naïve + LM and other Lcb + LM (ecadherin is the big one I noticed). Why would LcbWT + LM have more barrier disturbance than naïve + LM? What would WT Lcb be doing to the barrier? Is this noted in any other instances? It appears that most of the graphs show that barrier integrity is not drastically altered comparing naïve + LM and LcbLAP + LM. The main difference is the amount of adherent LM, which has already been covered in earlier figures.

10. Supp Figure 6: Shows that Lcb on its own (no LM infection) does not induce change in barrier integrity. This again brings up the question, what is happening during LcbWT + LM infection? Why is there such a difference in expression of barrier proteins?

11. Figure 6: IFN γ , IL-10, TGF β all increased with Lcb pretreatment (more when LAP present than when not.) CD11c $^{+}$ and FoxP3 $^{+}$ cells also increased. It is interesting that some of the cytokines were upregulated (IFN γ , IL-10, TGF β) even before LM infection. The presence of LcbLAP helped maintain these cytokines at higher levels during LM infection, whereas they decreased in naïve and LcbWT

upon infection with LM. What are the long term effects of higher levels of cells expressing these cytokines? How long would they stay upregulated with LcbLAP treatment?

12. Supp 8: Cytokine analysis after in vivo infection. Again, this demonstrates changing cytokine levels based on Listeria infection and Lcb pretreatment. It might have been helpful to have a more quantitative method of measuring cytokine production, such as flow cytometry, ELISA, or multiplex, instead of relying almost entirely on IHC/immunofluorescence.

13. Supp Figure 9: shows lack of increase of CD8+ T cells and CD4+ T cells during LM infection after treatment with LcbLAP. Is this lack of increase of CD8+ T cells mediated by the LcbLAP or by the decreased number of adherent LM? Could this be compared to Δ Lap LM infection?

14. While the authors demonstrated differences in cytokine production and expression, and gave rationale for each cytokine studied, there isn't a statement describing how the changes affect the end results. It's basically observing 'cytokine changes' as an answer, without much detail or explanation.

Minor point

1. Correct sentence beginning line 22: 'Here, we created bioengineered Lactobacillus probiotics (BLP) expressing the Listeria adhesion protein (LAP) on the surface of Lactobacillus casei isolated from a non-pathogenic Listeria (*L. innocua*) and a pathogenic Listeria (*Lm*)' to 'Here, we created bioengineered Lactobacillus probiotics (BLP) expressing the Listeria adhesion protein (LAP) isolated from a non-pathogenic Listeria (*L. innocua*) and a pathogenic Listeria (*Lm*) on the surface of Lactobacillus casei' for clarity.

Point-by-point Response to the Reviewers' Comments

NCOMMS-20-13603

Reviewer #1:

The manuscript by Drolia et al is a comprehensive study of the effects of engineered probiotics on *Listeria monocytogenes* pathogenesis. The work provides a number of new insights and advancements for the field. These include (1) the description of a novel approach for engineered probiotics; (2) insights into the likely mode(s)-of-action of such probiotics and (3) insights into *Listeria* pathogenesis. (4) As the engineered (BLP) strains also show enhanced colonization of the mouse gut the work also suggests an approach to enhance engraftment of probiotics in the gut through rational engineering.

Response: We appreciate the reviewer's recognition of the number of new advancements our study provides to the probiotic and *Lm*-pathogenesis field.

Main comments

1. In many cases the authors use both in vivo analyses supported by in vitro (CACO-2) studies to investigate mechanisms and the data are convincing. However, in other cases (e.g. the goblet cell work - Fig4) this is not possible & the authors rely on in vivo analysis alone. This results in a difficulty in explaining the phenomenon examined whereby BLP reduces both *Lm* numbers and loss of goblet cells. In this scenario it is not clear whether the preservation of goblet cells/barrier function during infection is a function of the activity of BLP on the host or a phenomenon related to lower numbers of *Lm* in the gut. I'm not sure it is possible to state (I309-310) that BLP 'limits *Lm*-induced loss of mucus-producing GCs' as BLP may actually be reducing numbers of *Lm* in the gut by other mechanisms and it is the resulting lower numbers of *Lm* that reduce the phenomenon of GC loss. I am suggesting that the authors address this in the language used, to introduce this caveat into a revised text.

This is also true for the data in Fig 5 - though the authors conclusions are better supported here by in vitro studies. Still I feel it is an issue that should be addressed by a statement in the revised text.

Response: We appreciate the suggestions. We agree. We cannot rule out that BLP-mediated preservation of goblet cells may be due to the lower burdens of *Lm* in the intestine. We have now revised our statement accordingly in the results (see lines 358- 362 and 412-415, revised Fig 6 and 7) and the discussion section (see lines 545-553).

2. Much of the histology work is supported by quantitative data/scores (e.g. in Fig 4). However, where there is no quantitation it would be useful to see some indication of number of fields that were examined per sample in order for the reader to appreciate how robust is the evidence for particular phenomena. This applies in particular to the studies of co-localization/co-aggregation. apologies if I have missed it in the text, but I feel that it should be made more obvious to the reader and quantified as much as possible.

Response: As suggested, we have quantified (Supplementary fig. 3d) the colocalization results (now Fig. 4e, f and Supplementary fig. 4e, f) and expressed the

data as the total no. of BLP cells colocalized with Hsp60 from 10 different high magnification fields in each of the three independent experiments (total 30 fields) that we performed.

We have also quantified the immunomagnetic bead captured co-aggregates (Fig. 4c) and expressed these results as the capture efficiency of *L. casei* cells with IMB-Lm complex from 10 different high magnification fields in each of the three independent experiments (total 30 fields). Additionally, the coaggregation results (IMB-Lm complex capture assay) have already been quantified with plating (CFU's, Fig. 4a) and now with our newly acquired data via crystal-violet biofilm assay (Supplementary Fig. 4b).

3. A strength of the work is that the BLP strains demonstrate better colonisation of the gut than the wild-type strain. Whilst I appreciate that this is a mouse study, the data may be useful for enhancing engraftment of probiotics in humans - as this is a significant issue in the field of probiotic research (where many strains are seen to be transient in the gut & therefore ineffectual). On the other hand the genetic manipulation of gut commensals (as opposed to *L. lactis*, which doesn't colonise the gut) is controversial as such strains may colonise for the life of the host leading to issues of GMM release etc. Whilst the Introduction to the paper mentions one of these issues I feel they could be expanded in the Discussion slightly to contextualise the work. **Whilst the paper is very well written I feel that the Discussion reads slightly as a re-iteration of the Results section (which is already very comprehensive). I feel that the authors should better emphasize some of the biotech implications of the work in the Discussion.**

L45-46: related to the above point the authors also suggest in the Introduction that it is proposed that probiotics compete with pathogens for adhesion sites - and a citation is given to a review article. However, I am not sure that good evidence exists for this. **So a strength of the current work is support for this concept & greater general discussion of this point would add to the quality of the Discussion section.**

Response: We appreciate the reviewer's recognition of the strengths of the study and suggestions to improve the discussion section. In the revised manuscript, we have revised the discussion section thoroughly and avoided reiteration of results as much as possible, and addressed the key issues as pointed by the reviewer: (i) Usefulness of the BLP strain for enhancing engraftment of probiotics in humans (lines 581-585); (ii) Life-long colonization and biocontainment of bioengineered probiotics have been addressed by creating an auxotroph mutant (lines 585-592). Matter of fact, we are fully aware of this biocontainment issue, which we are now in the process of addressing (a proposal has been submitted to secure funding); (iii) Claim for direct demonstration of competition of pathogen adhesion site (receptor) has been made (lines 518-529).

4. Minor comments:

The following sentences do not scan particularly well and could be revised for clarity:

L59-63

L126-129

L421-424
L449-452
L501-503

Response: We have revised these sentences for clarity. These sentences are highlighted, and the revised lines are:

Lines 60-64

Lines 126-132

Lines 475-479

Lines 449-452 has been deleted while revising the reiterations in the discussion section.

Lines 540-541

5. Did the authors measure plasmid retention in the engineered strains?

Response: The plasmid contains the gene for erythromycin. All BLP cell enumeration was done by plating samples on MRS plates containing erythromycin (2 µg/ml) and vancomycin (300 µg/ml) to ensure plasmid retention. Furthermore, we have also verified the expression of LAP in BLP isolated from feces from day 10 (**Supplementary Fig. 2f**), again demonstrating that the plasmid is maintained in the BLP strains during colonization in the gut. In our new experiment (**Fig 3**), we also used the same antibiotics for isolation of BLP strain 12 days post-feeding, again providing indirect evidence for maintenance of the plasmid in the BLP strain.

6. Ref 26. To my knowledge there is an updated review from this group that would be more appropriate here: Front Cell Infect Microbiol. 2014 Feb 5;4:9.

Response: The suggested updated review reference (27) is added. It was our oversight.

Reviewer #2:

Drolia et al have followed up on their previous work which identified LAP as an additional receptor for *Listeria monocytogenes* (Lm) entry into the intestine, and applied this finding to engineering a probiotic strain of *Lactobacillus* that expresses the LAP receptors from either Lm or *Listeria innocua*. **The paper is well done and impactful, in particular for the probiotic field in which it has been a challenge to establish colonization that would be lasting and effective. I have a couple of suggestions that I think would broaden the manuscript's impact and some areas that would require additional experiments to substantiate the authors' conclusions.**

Response: We appreciate the reviewer's recognition of the impact of our study on the probiotic field. As suggested, we have performed additional experiments with a large number of animals (n=43) to broaden the manuscript's impact. Please see below:

Major experimental suggestions:

1. The probiotic and anti-Listerial effects have been extensively characterized in two experimental models that are particularly permissive to *Listeria* infection (Caco-2 cells and A/J mice). **The authors performed a very comprehensive set of experiments to characterize that BLP strains prevent Lm infection, downstream signaling, loss of barrier integrity, increased goblet cell proliferation and immune responses.** It would extend and generalize the findings of the manuscript to show the same effect on Lm CFUs in C57B6 mice and/or in a less permissive cell line for infection that can form better tight junctions (i.e. the cell line that has been used historically by many groups to assess entry phenotypes **MDCK cells** which are known to create extremely tight epithelial barriers that replicate the situation in vivo). **It would be excessive to ask for all of the experiments to be replicated but assessing bacterial adhesion and invasion in less permissive cells lines or mouse models would be sufficient to increase the potential scope of the findings in a more stringent biological system.**

Response: As suggested, we verified the protective effect of BLP using the MDCK cell line. In the interest of time, we could not use the C57BL/6 mice to show the protective effect. Consistent with our Caco-2 cell data (**Fig. 1k-m**), pretreatment of BLP strains significantly reduced (~90-99 % reduction) *Lm* adhesion (**Supplementary Fig. 1d**), invasion (**Supplementary Fig. 1e**), and translocation (**Supplementary Fig. 1f**) across MDCK monolayers. These data validate unequivocally the protective effects of the BLP strains even in a less-permissive cell model.

2. The authors hypothesize that the BLP strains occupy HSP60 and thus outcompete Lm binding as one potential mechanism of entry however in principle you would still get binding to E-cadherin and C-met that would allow entry to the tips of villi and near goblet cells under these conditions. If HSP60 binding is occluded the authors could demonstrate that in Caco2 cells using an HSP60 antibody prior to Lm infection as a second experimental demonstration of their hypothesis.

Response: To further confirm the contribution of host Hsp60 in *Lm* interaction with intestinal epithelial cells, as suggested, we pretreated Caco-2 cells with an Hsp60-specific antibody prior to *Lm* exposure, which significantly reduced the adhesion (~10-fold), invasion (~3-fold), and translocation (~12-fold) of the *Lm* WT strain (**Supplementary Fig. 1m-o**).

3. Based on their data, I was more interested in the chains of BLPs that form and seem to trap or bind to *Lm* in the mucus and the intestinal lumen. Does this binding occur using a biofilm assay in in vitro planktonic co-culture between BLP and *Listeria monocytogenes* with and without LAP (e.g. crystal violet binding to test tubes as is the case for ActA)? Are the chains formed in the intestine or normally during growth by virtue of LAP overexpression? The microscopy seems to indicate longer chains for BLPs with LAP than LbcWT but that could be based on the one representative image that is shown. Would other gastroenteric pathogens like *Salmonella* be trapped as well physically and restricted from entry or would it be unique to *Lm* cell-wall binding of LAP?

Response: We believe the chain formation is linked to biofilm formation due to the expression of LAP. As suggested by the reviewer, we checked the biofilm-forming ability of BLP by crystal violet staining and data clearly show several-fold higher biofilm formation by BLP than the LbcWT on a plastic surface in both monoculture or co-culture with *Lm* (**Supplementary Fig. 4b**). Interestingly, we did not observe any discernible biofilm formation when BLP strains were co-cultured with *Salmonella enterica* serovar Typhimurium (**Supplementary Fig. 4b**). Incidentally, these BLP strains were also unable to prevent adhesion of *S. Typhimurium* and *E. coli* O157:H7 to Caco-2 cells in a competitive exclusion assay while they were successful in preventing adhesion of *Lm* (an unpublished observation made by another researcher who is not part of the current study). Nevertheless, these data indicate that the LAP expression on the surface of BLP promotes biofilm formation (verified by both *in vitro* and *in vivo* exp) and competitive exclusion is limited to *Lm* since the adhesion of two other enteric pathogens was not interfered by BLP.

4. The timeline of colonization in the mouse model means that there would have still been BLPs in the system when *Lm* is administered. **Would there still be BLP colonization following a longer gap prior to infection say one or two weeks? More specifically how lasting is the BLP colonization and the protective effect it confers, or would it require continuous BLP administration?** This gets at the question of colonization of the mucosa versus trapping and clearing *Lm* through the chains and could also be assessed by *Lm* shedding in the feces over time (beyond the 24 and 48 hours shown) and how that is altered by BLPs.

Response: It is indeed an intriguing question which we addressed with a new animal experiment (see new Fig 3). We examined colonization and persistence (beyond 10 days of probiotic feeding) of BLP for additional 12 days (total length of study, 22 days, Fig 3a) and the nature of protection against *Lm* during this period. Fecal shedding (indicator of intestinal colonization) data clearly indicated significantly higher colonization and persistence of BLP until the end of the study while LbcWT was undetectable from day 14 to 20 (Fig 3b). Analysis of intestinal content (after necropsy) showed a similar trend (Fig 3c).

Protection against *Lm* infection (post-probiotic feeding) was analyzed by enumerating *Lm* in the intestinal and extraintestinal tissues on days 12, 17, and 22 (**Fig 3a**). The highest protection was observed on days 12 and 17 i.e., 2-7 days after probiotic feeding was stopped. On day 22 (i.e., 12 days after feeding) the protection was evident but significantly lower than day 17 (**Fig 3d-3j**). These data indicate prolonged persistence of the BLP strain at least for additional 12 days in the mouse gut after probiotic feeding was stopped with the capacity to protect the host from listeriosis during this period. The time and funding constraints prevented us from continuing this study any further. Moreover, in our mouse survival experiment (**Fig 2p**), 92% BLP-fed mice survived 10 days post BLP feeding while only 60% survived after LbcWT feeding again providing circumstantial evidence for BLP persistence and continued protection against listeriosis.

Based on these new data, we believe BLP interaction with epithelial cells and biofilm formation are major factors for prolonged BLP persistence on the epithelial surface and for competitive exclusion of *Lm* after the probiotic feeding has been stopped. Based on these data, daily BLP feeding may not be necessary, but once a week may be sufficient to achieve the protective effect.

5. Also, why would there be less *Lm* shed if you inoculate the same amount and get less infection? Wouldn't it follow that the bacteria would be more shed into the feces originally? Do you need to add a Day 1 time point to see the majority of the *Lm* that is shed into the feces? Where else would it go?

Response: This is an interesting observation and we concur with reviewer's speculation for possible early shedding of *Lm* after BLP feeding. As suggested, in our new animal experiment, we analyzed the fecal shedding of *Lm* at 12, 24 and 48 hpi, and the data indeed show significantly increased *Lm* shedding at 12 hpi which gradually decreased at 24 and 48 hpi (**Supplementary Fig. 2i**) (see lines 210-210).

Minor suggestions:

6. Fractionation controls are missing– for both the HSP60 localization and the Listeria/BLP cell wall fractionation experiments the fractions should be run on the same blot with controls that indicate that the fractionation worked (i.e. for Listeria InIA, not sure what would be appropriate for Lbc) or a membrane/TJ marker for Caco2. Since the localization of LAP on *L.innocua* is discussed it should be included in the fractionation experiments to show that it does not attach to the cell wall as it does for *Lm*. This was shown with fluorescent imaging but should be demonstrated with WB like it is for *Lm*. In addition, if it is secreted and released this could be shown through a TCA precipitated secreted fraction a control for this could be p60.

Response: We have repeated immunoblot analysis for Hsp60 localization (**Supplementary Fig.1k**) and included appropriate fractionation controls. Briefly, occludin was used as a Caco-2 membrane marker and MEK-1/2 as a cytosolic marker, which demonstrated the absence of detectable MEK-1/2 levels in the membrane fraction and the absence of detectable occludin levels in the cytosolic fraction, confirming the fractionation was successful.

Additionally, we have repeated our Immunoblot analysis of protein preparations from the bacterial cell wall and supernatant (TCA precipitation, **Fig. 1b**) confirming complementation, and cell wall expression and secretion of *L. innocua* LAP in the *lap*⁻ mutant strain (*lap*⁻+*lap*^{Lin}) and of *Lm* LAP in the *lap*⁻ mutant strain (*lap*⁻+*lap*^{Lm}, right panel).

As suggested, we have also included *L. innocua* as control and consistent with our previous study (Jagadeesan et al 2010), no cell wall association or secretion of LAP was observed in *L. innocua* strain (**Fig. 1b**). Also, we have used InIA and N-acetylmuramidase (NamA) as a *Lm* cell-wall protein and secreted protein fractionation controls, respectively. Furthermore, Coomassie-stained gel (bottom panel) was used to confirm equal loading. Instead of P60, we used NamA as a *Lm* secreted protein marker.

As for *Lbc*, we have demonstrated the cell-wall expression of LAP in the BLP strains via immunofluorescence imaging (**Fig. 1g**) and immunoblots (**Fig. 1f**). Since the antibody against the cell wall of *Lbc* (ATCC 334) is unavailable commercially, we were unable to use them as controls. However, a similar cellular fractionation technique was used to isolate the cell-wall protein fractions from *Lbc*, suggesting our fractionation technique is effective.

7. The histology used throughout is very nice but it would be good to have zoomed out images that show the gross distribution of *Lm* and BLP to demonstrate distribution throughout the intestine and show that infected villi were not specifically sought out.

Response: As suggested, we have presented zoomed out images for **Fig 4g** in **Supplementary Fig 4a** for improved assessment and visibility of *Lm* and BLP distribution throughout the intestine.

8. Additionally, the authors undertook a heroic effort in quantifying immune cells by histology but wouldn't FACS be more quantitative and representative of immune infiltrate into the total tissue?

Response: We agree that FACS would have been more quantitative and less time-consuming; however, given the number of tissue samples from 8 treatment groups has to be analyzed within 24 h of animal sacrifice by FACS would be hugely cumbersome. Moreover, we had to perform bacterial tissue burden analysis at the same time too. Therefore, we chose histology which allowed us to perform these experiments in fixed tissues over several days/months. Furthermore, histology also provided visual evidence of the location and distribution of target cells throughout the villi and crypt. Moreover, this approach is now being used by others for relative quantitative measurements of immune infiltrates (Formentini et al 2017, Nalle et al 2019).

Reviewer #3:

The manuscript by Drolia et describes that the pre-colonization of *Listeria monocytogenes* (LM)-susceptible A/J mice with *Lactobacillus casei* expressing the LM surface protein LAP (LcbLAP) probiotics decreases the severity of LM infections. The authors report lower numbers of invasive bacteria in the gut and distal organs, as well as improved clinical scores and survival. The effects of LcbLAP are attributed to three main mechanisms: (1) competitive exclusion of LM adhesion to epithelial cells; (2) improved intestinal barrier function; and (3) contact-dependent immune-modulation.

The use of probiotics to prevent or mitigate LM infection has been of recent interest, however most of these studies do not utilize LM protein-producing engineered probiotic strains. Additionally, these studies tend to focus on the production of bacteriocins and other competitive exclusion mechanisms, whereas the present study outlines multiple functions of the engineered LcbLAP strain: (1) competition, (2) barrier integrity, and (3) immuno-modulation. A previously published paper regarding the use of LcpLAP in preventing LM infection looked at adhesion, invasion, and translocation, as well as probiotic induced anti-LM cytotoxicity. The earlier strain did not show as promising results for prevention of dissemination of LM infection. This paper looked at potential mechanisms of LcbLAP for mitigating lethal LM infection.

Overall, the data presented appears to convincingly support the premise that this engineered probiotic strain can help reduce severe LM infection. What is unclear is the actual utility of this approach – it would appear that the probiotic may need to be continually consumed to offer protection, and it seems unlikely that probiotics would have much effect after LM has successfully crossed the intestinal barrier, and therefore are unlikely to represent a treatment (vs a preventative). This caveat might be mitigated if it was demonstrated that the probiotic provided some protection when not consumed daily – if it was able to stably colonize the GI tract.

Response: We appreciate the reviewer's recognition of the importance of our study to the probiotic field. As suggested, we have performed additional experiments with a large number of animals to broaden the manuscript's impact and the utility of our approach, especially prolonged colonization/persistence and consequent protection against *Lm* infection (**see the response to comments # 4, Reviewer 2 and Fig. 3**). We have also clearly articulated in the manuscript that the BLP generated in this study is ideal for prevention and based on the new data, it is not necessary to consume the probiotic daily to receive the benefit, once a week may be sufficient (**see the response to comments # 4, Reviewer 2**). For therapeutic application, additional animal experiments need to be performed, but based on our cell culture experiment, BLP may have limited utility as therapeutics.

Specific comments:

1. Figure 1: Data from Figure 1 was obtained using standard techniques to monitor adhesion, invasion, and translocation of bacteria. Data are presented as mean +/- SEM; statistical analysis (one or two way ANOVA with Tukey's multiple comparison) is appropriate. Adhesion/invasion/translocation of Lap- LM after LcbLAP treatment would be a nice control to compare how important Lap interaction is in preventing adhesion/invasion/translocation.

Response: As suggested, we have repeated the experiment using the *lap*⁻ strain. As expected, BLP was able to inhibit both LmWT and the *lap*⁻ strain interaction with Caco-2 cells, albeit the effect, was more evident against WT since it interacts more than the *lap*⁻ strain (**Supplementary Fig. 1g-i, Line 152-156**). These data suggest that the Lm LAP interaction with intestinal epithelial cells is crucial for BLP-mediated exclusion of Lm.

2. Line 136: a 100% increase in adhesion is essentially a two-fold increase.

Response: We have revised as suggested.

3. Figure 2: CFU data shown as mean +/- SEM. The median is less affected by outliers vs SEM and is preferable. How stable is LcbLATPLm colonization? Does protection require continuous feeding of the probiotic?

Response: We agree with the reviewer and have presented all CFU data as *median +/- interquartile range* in **Fig. 2** and **3** in the revised manuscript (line 1308-1311 and 1353-1357). For BLP colonization and protection against infection when probiotic feeding is stopped comment, please see the response to **comments # 4, Reviewer 2 and Fig. 3**.

4. It is unclear if statistical significance is measured between each group individually, or clumped together (ie naïve vs LcbWt vs LcbLAPLM vs LcbLAPlim or naïve + LcbWT vs LcbLAPLM + LcbLAPlim). Statistical tests used are appropriate, but data being compared could be clarified.

Response: Statistical significance was determined by using the Mann-Whitney nonparametric test (**Fig. 2 b, d-h, k-o and Fig.3b-j**) and comparisons were made between each treatment group individually. We have revised the figure legends to clarify this in the text (line 1309-1311 and 1353-1357)

5. Supp Figure 2: Is part H necessary? It is hard to distinguish major differences between mice (some ruffling is evident, but weight loss, breathing changes, and movement can't be discerned from photographs).

Response: We agree with the reviewer and the figure is now omitted.

6. Figure 3: Anti-LM microbeads were used to capture LM and co-capture Lcb. Is it possible that the anti-LM immunobeads could recognize/capture LM-LAP protein expressed on Lcb? This experiment demonstrated relatively equal numbers of LM and LcbLAP captures. Immunofluorescence and IHC show colocalization of Lcb and LM and Lcb and Lin in vivo.

Response: Our data clearly show, very little non-specific capture of Lbc or LAP-expressing Lbc by the immunomagnetic beads (see first three bars in **Fig 4a** and **Supplementary Fig. 3a**).

7. Supp Figure 3: Colocalization of Lcb with Hsp60. Anti-Hsp60 antibody inhibition of binding by LcbLAP is a nice touch. Immunoprecipitation of LcbLAP with Hsp60 might also be valuable to show?

Response: We have demonstrated the interaction of Lcb through immunofluorescence imaging (**Fig. 4e, f and Supplementary Fig. 3e, f**) and Hsp60 knock-down and overexpression studies (**Fig. 1 n**). During this review, we have also conducted another experiment where we have shown that pretreatment of Caco-2 cells with anti-Hsp60 antibody can block *Lm* adhesion, invasion, and translocation (**Supplementary Fig. 1 m, n, o**). While immunoprecipitation may provide another evidence, we believe these data unequivocally confirm LAP-Hsp60 interaction during BLP-treatments.

8. Supp Figure 4: Alcian blue staining shows change in number of goblet cells. This is enumerated by graph. It would be nice to see if there is statistical significance between LcbLAP with and without infection.

Response: Statistical significance has been added (**Supplementary Fig. 5a**). No significant difference is observed in goblet cell counts after BLP treatments with or without *Lm* infection.

9. Fig. 5E: Cell membrane expression of junction proteins in LcbWT + LM looks very different from naïve + LM and other Lcb + LM (ecadherin is the big one I noticed). Why would LcbWT + LM have more barrier disturbance than naïve + LM? What would WT Lcb be doing to the barrier? Is this noted in any other instances? It appears that most of the graphs show that barrier integrity is not drastically altered comparing naïve + LM and LcbLAP + LM. The main difference is the amount of adherent LM, which has already been covered in earlier figures.

Response: It was our oversight. In the revised manuscript, we have included a picture that is representative (**Figure 6e and Supplementary Fig. 6c**). *Lm* alone and LcbWT + *Lm* have similar barrier disturbance (claudin-1, occludin and E-cadherin, **Figure 6e and Supplementary Fig. 6c**) and quantification analysis of these groups clearly suggest that they are not statistically different (**Sup. Fig. 6b**).

Regarding, Naïve + *Lm* and LcbLAP + *Lm* comparisons, there may have been confusion. We have observed a drastic difference between these treatment groups. The cell-cell junctions (claudin-1, occludin and E-cadherin) are intact in LcbLAP+*Lm* (very few intracellular puncta) but not in Naïve +*Lm* (very high intracellular puncta) (**Fig. 6 e-f and Supplementary. Fig. 6b, c and f**).

We cannot rule out that BLP-mediated preservation of barrier integrity may also be due to the lower burdens of *Lm*. As suggested by **reviewer 1 comment 1**, we have now revised our statement accordingly in the results (see lines 358- 362 and 412-415, revised Fig 6 and 7) and the discussion section (see lines 545-553).

10. Supp Figure 6: Shows that Lcb on its own (no LM infection) does not induce change in barrier integrity. This again brings up the question, what is happening during LcbWT + LM infection? Why is there such a difference in expression of barrier proteins?

Response: Please see the response to comment # 9. Briefly, LbcWT + *Lm* has similar barrier disturbance as *Lm* alone and these groups are not statistically different (Fig, 6e, f and Supplementary Fig. 6b, c and f).

11. Figure 6: IFN, IL-10, TGF β all increased with Lcb pretreatment (more when LAP present than when not.) CD11c+ and FoxP3+ cells also increased. It is interesting that some of the cytokines were upregulated (IFN γ , IL-10, TGF β) even before LM infection. The presence of LcbLAP helped maintain these cytokines at higher levels during LM infection, whereas they decreased in naïve and LcbWT upon infection with LM. What are the long term effects of higher levels of cells expressing these cytokines? How long would they stay upregulated with LcbLAP treatment?

Response: This is an intriguing and interesting comment. We speculate eventual immunological homeostasis promoting improved gut health during prolonged BLP feeding. Because of the time and funding constraints, we could not accommodate this study at this time, but it will be addressed in the future.

12. Supp 8: Cytokine analysis after in vivo infection. Again, this demonstrates changing cytokine levels based on *Listeria* infection and Lcb pretreatment. It might have been helpful to have a more quantitative method of measuring cytokine production, such as flow cytometry, ELISA, or multiplex, instead of relying almost entirely on IHC/immunofluorescence.

Response: We have used ELISA to quantify the levels of TNF α (Fig. 7e), IL-6 (Fig. 7f), and IFN γ (Fig. 7g). Immunohistochemistry (IHC) was used for quantifying IL-10 and TGF β since ELISA kits used in this study failed to accurately quantify the levels of IL-10 even after several attempts. IHC has been proven to be useful for quantitative measurements (Formentini et al 2017, Kanda et al 2016, Ran-Ressler et al).

13. Supp Figure 9: shows lack of increase of CD8+ T cells and CD4+ T cells during LM infection after treatment with LcbLAP. Is this lack of increase of CD8+ T cells mediated by the LcbLAP or by the decreased number of adherent LM? Could this be compared to Δ Lap LM infection?

Response: Subset analyses of T-cells showed fewer CD8+ T cells in BLP-treated mice 48 h-post *Lm*-challenge (now Supplementary Fig. 10a) while significantly increased numbers of CD4+ T cells (~66%, now Supplementary Fig. 10b), relative to naïve or LbcWT-treated mice pre- or 48 h-post *Lm*-challenge. We cannot rule out that the reduced levels of adherent *Lm* in BLP-treated mice may have also contributed to the lack of an increase of CD8+ T cells. However, our regret for not being able to accommodate this experiment at this time since we focused more on the long-term-persistence and protection afforded by the BLP (Fig. 3).

14. While the authors demonstrated differences in cytokine production and expression, and gave rationale for each cytokine studied, there isn't a statement describing how the changes affect the end results. It's basically observing 'cytokine changes' as an answer, without much detail or explanation.

Response: We have revised the text to provide more explanation. Briefly, our results suggest that oral administration of BLP promotes the production of IFN γ (Fig. 7g) for

effective *Lm* clearance and upregulates IL-10 (Fig. 7h and Supplementary Fig. 9a) and TGF- β (Fig. 7i and Supplementary Fig. 9b) that prevent excessive inflammation, consistent with reduced histopathology and inflammation in these mice (Fig. 5a), thus maintaining intestinal immune homeostasis (See lines 452-455).

Minor point

1. Correct sentence beginning line 22: 'Here, we created bioengineered *Lactobacillus* probiotics (BLP) expressing the *Listeria* adhesion protein (LAP) on the surface of *Lactobacillus casei* isolated from a non-pathogenic *Listeria* (*L. innocua*) and a pathogenic *Listeria* (*Lm*)' to 'Here, we created bioengineered *Lactobacillus* probiotics (BLP) expressing the *Listeria* adhesion protein (LAP) isolated from a non-pathogenic *Listeria* (*L. innocua*) and a pathogenic *Listeria* (*Lm*) on the surface of *Lactobacillus casei*' for clarity.

Response: We have revised this statement accordingly. See lines 22-25

References

Formentini, Laura, et al. "Mitochondrial ROS production protects the intestine from inflammation through functional M2 macrophage polarization." *Cell Reports* 19.6 (2017): 1202-1213.

Jagadeesan, Balamurugan, et al. "LAP, an alcohol acetaldehyde dehydrogenase enzyme in *Listeria*, promotes bacterial adhesion to enterocyte-like Caco-2 cells only in pathogenic species." *Microbiology* 156.9 (2010): 2782-2795.

Kanda, Toshihiro, et al. "Enterococcus durans TN-3 induces regulatory T cells and suppresses the development of dextran sulfate sodium (DSS)-induced experimental colitis." *PLoS One* 11.7 (2016): e0159705.

Nalle, Sam C., et al. "Graft-versus-host disease propagation depends on increased intestinal epithelial tight junction permeability." *The Journal of clinical investigation* 129.2 (2019): 902-914.

Ran-Ressler, Rinat R., et al. "Branched chain fatty acids reduce the incidence of necrotizing enterocolitis and alter gastrointestinal microbial ecology in a neonatal rat model." *PloS one* 6.12 (2011): e29032.

REVIEWERS' COMMENTS

Reviewer #1 (Remarks to the Author):

The revised manuscript by Drolia and coworkers represents a significant advance in the concept of engineered probiotics/delivery vectors. The authors have made significant efforts to answer the queries of the reviewers. I have no further comments.

Reviewer #2 (Remarks to the Author):

Drolia and colleagues did a significant amount of work to improve and revise the manuscript. I think the manuscript is impactful and will be of broad interest to the field.

My suggestions and concerns have been addressed.

Reviewer #3 (Remarks to the Author):

The authors have done an admirable job of addressing concerns raised by reviewers during the previous review. The result is an impressive body of work that should be of interest to a broad community interested in the potential impact of probiotics on intestinal health.

Point-by-point response to the Reviewers' comments

NCOMMS-20-13603

REVIEWER COMMENTS

Reviewer #1 (Remarks to the Author):

The revised manuscript by Drolia and coworkers represents a significant advance in the concept of engineered probiotics/delivery vectors. The authors have made significant efforts to answer the queries of the reviewers. I have no further comments.

Response: We appreciate the reviewer's valuable suggestions to improve the manuscript. We take this opportunity to thank the reviewer for recognizing the significance of engineered probiotics/delivery vectors.

Reviewer #2 (Remarks to the Author):

Drolia and colleagues did a significant amount of work to improve and revise the manuscript. I think the manuscript is impactful and will be of broad interest to the field.

My suggestions and concerns have been addressed.

Response: We thank the reviewer for the constructive and critical comments that aided us to broaden and improve our study. We certainly appreciate the reviewer's positive view.

Reviewer #3 (Remarks to the Author):

The authors have done an admirable job of addressing concerns raised by reviewers during the previous review. The result is an impressive body of work that should be of interest to a broad community interested in the potential impact of probiotics on intestinal health.

Response: We thank the reviewer for providing suggestions that significantly improved our manuscript. We appreciate the reviewer's recognition of our efforts and the importance of our study to the probiotic field.